# Energy-Efficient Scheduling with Predictions

**Eric Balkanski**
Columbia University
eb3224@columbia.edu

**Noemie Perivier**
Columbia University
np2708@columbia.edu

**Clifford Stein**
Columbia University
cliff@ieor.columbia.edu

**Hao-Ting Wei**
Columbia University
hw2738@columbia.edu

## Abstract

An important goal of modern scheduling systems is to efficiently manage power usage. In energy-efficient scheduling, the operating system controls the speed at which a machine is processing jobs with the dual objective of minimizing energy consumption and optimizing the quality of service cost of the resulting schedule. Since machine-learned predictions about future requests can often be learned from historical data, a recent line of work on learning-augmented algorithms aims to achieve improved performance guarantees by leveraging predictions. In particular, for energy-efficient scheduling, Bamas et. al. [NeurIPS '20] and Antoniadis et. al. [SWAT '22] designed algorithms with predictions for the energy minimization with deadlines problem and achieved an improved competitive ratio when the prediction error is small while also maintaining worst-case bounds even when the prediction error is arbitrarily large.

In this paper, we consider a general setting for energy-efficient scheduling and provide a flexible learning-augmented algorithmic framework that takes as input an offline and an online algorithm for the desired energy-efficient scheduling problem. We show that, when the prediction error is small, this framework gives improved competitive ratios for many different energy-efficient scheduling problems, including energy minimization with deadlines, while also maintaining a bounded competitive ratio regardless of the prediction error. Finally, we empirically demonstrate that this framework achieves an improved performance on real and synthetic datasets.

## 1 Introduction

Large data centers and machine learning models are important contributors to the growing impact that computing systems have on climate change. An important goal is thus to efficiently manage power usage in order to not only complete computing tasks in a timely manner but to also minimize energy consumption. In many operating systems, this tradeoff can be controlled by carefully scaling the speed at which jobs run. An extensive area of scheduling has studied such online (and offline) speed scaling problems (see, e.g., [1]). Since the speed of many processors is approximately the cube root of their power [25, 15], these works assume that the power of a processor is equal to speed to some power $\alpha \geq 1$, where $\alpha$ is thought of as being approximately 3 [28, 10] and the total energy consumption is power integrated over time.

Online energy-efficient scheduling algorithms have mostly been evaluated using competitive analysis, which provides robust guarantees that hold for any instance. However, since competitive analysis evaluates algorithms over worst-case instances, it can often be pessimistic. In particular, it ignores the

37th Conference on Neural Information Processing Systems (NeurIPS 2023).

| Problem | Previous results | | Our results with predictions | |
|---|---|---|---|---|
| | without predictions | with predictions | Consistency | Robustness |
| Flow time | 2 [3] | None | $(1 + (2\lambda)^{\frac{1}{\alpha}})^{\alpha}$ | $\frac{1 + 2^{2\alpha+1}\lambda^{\frac{1}{\alpha}}}{\lambda}$ |
| Fractional weighted flow time | 2 [14] | | | |
| Integral weighted flow time | $O((\frac{\alpha}{\log \alpha})^2)$ [12] | | $(1 + ((\frac{\alpha}{\log \alpha})^2\lambda)^{\frac{1}{\alpha}})^{\alpha}$ | $\frac{1 + (\frac{\alpha}{\log \alpha})^2 2^{2\alpha}\lambda^{\frac{1}{\alpha}}}{\lambda}$ |
| Deadlines | $e^{\alpha}$ [10] | [7, 4] | $1 + \lambda$ | $O(\frac{4^{\alpha^2}}{\lambda^{\alpha-1}})$ |

Table 1: The best-known competitive ratios for 4 energy-efficient scheduling problems, previous work studying these problems in the algorithms with predictions framework, and our consistency and robustness results, for any $\lambda \in (0, 1]$. Note that when $\lambda$ is sufficiently small, the consistency improves over the best-known competitive ratios, while also maintaining bounded robustness. A detailed comparison with the results of [7, 4] for deadlines is provided in Section 1.2.

fact that, in the context of scheduling, future computation requests to computing systems can often be estimated from historical data. A recent line of work on algorithms with predictions aims to address this limitation by assuming that the algorithm designer is given access to machine-learned predictions about the input. In the context of online algorithms, where this line of work has been particularly active, the predictions are about future requests and the goal is to achieve improved competitive ratios when the predictions are accurate (consistency), while also maintaining the benefits of worst-case analysis with guarantees that hold even when the predictions are arbitrarily wrong (robustness).

In this framework with predictions, Bamas et al. [7] and Antoniadis et al. [4] recently studied the energy minimization with deadlines problem, which is a classical setting for energy-efficient scheduling (see, e.g., [28, 10]). However, there are many scenarios where the jobs do not have strict deadlines and the goal is instead to minimize the job response time. In energy plus flow time minimization problems, which are another family of energy-efficient scheduling problems that have been extensively studied in the setting without predictions, the objective is to minimize a combination of the energy consumption and the flow time of the jobs, which is the difference between their release date and completion time (see, e.g., [2, 12, 14, 3]).

In this paper, we study a general energy-efficient scheduling problem that we augment with predictions. This general problem includes both energy minimization with deadlines, which has been previously studied with predictions, and energy plus flow time minimization, which has not been previously studied with predictions, as well as many other variants and generalizations. In particular, the flow time problem with predictions introduces challenges that require novel learning-augmented scheduling algorithms (see Section 3 for additional discussion).

## 1.1 Our results

An instance of the General Energy-efficient Scheduling (GES) problem is described by a collection $\mathcal{J}$ of $n$ jobs and an arbitrary quality of service cost function $F$. Each job $(j, r_j, p_j) \in \mathcal{J}$ consists of a release time $r_j$, a processing time $p_j$, and an identifier $j$ (and potentially other parameters such as weights $v_j$ or deadlines $d_j$). A schedule $S$ is specified by the speeds $s_j(t)$ at which job $j$ is processed by the machine at time $t$. The goal is to find a schedule of minimum cost $E(S) + F(S, \mathcal{J})$, where the energy consumption of a schedule is $E(S) = \int_{t \geq 0} (\sum_j s_j(t))^{\alpha} dt$, for some constant $\alpha > 0$. In the general energy-efficient scheduling with predictions (GESP) problem, the algorithm is given at time $t = 0$ a collection $\hat{\mathcal{J}}$ of $\hat{n}$ predicted jobs $(j, \hat{r}_j, \hat{p}_j)$, which is a similar prediction model as in [7]. For all our results, we assume that the quality cost function $F$ is monotone and subadditive, which are two mild conditions that are satisfied for the problems with flow times and with deadlines.

**Near-optimal consistency and bounded robustness.** Our first goal is to design an algorithm for the GESP problem that achieves a good tradeoff between its consistency (competitive ratio when the predictions are exactly correct) and robustness (competitive ratio when the predictions are arbitrarily wrong). Our first main result is that for any instance of the GES problem for which there exists a constant competitive algorithm and an optimal offline algorithm, there is an algorithm with predictions that is $1 + \epsilon$ consistent and $O(1)$ robust for any constant $\epsilon \in (0, 1]$ (Corollary 3.5). Since problems

with the flow time and the problem with deadlines admit constant-competitive algorithms, we achieve a consistency that is arbitrarily close to optimal while also maintaining constant robustness for these problems (see Table 1 for a summary of problem-specific upper bounds). We complement this result by showing that there is a necessary trade-off between consistency and robustness for the flow time problem: for any $\lambda > 0$, there is no $1 + \lambda$-consistent algorithm that is $o(\sqrt{1 + 1/\lambda})$-robust (Appendix A.2).

**The competitive ratio as a function of the prediction error.** The second main result is that our algorithm achieves a competitive ratio that smoothly interpolates from the $1 + \epsilon$ consistency to the constant robustness as a function of the prediction error (Theorem 3.4). To define the prediction error, we denote by $\mathcal{J}^+ = \mathcal{J} \cap \hat{\mathcal{J}}$ the jobs that are correctly predicted. We define the prediction error $\eta = \frac{1}{\mathsf{OPT}(\hat{\mathcal{J}})} \max\{\mathsf{OPT}(\mathcal{J} \setminus \mathcal{J}^+), \mathsf{OPT}(\hat{\mathcal{J}} \setminus \mathcal{J}^+)\}$, which is the maximum between the optimal cost of scheduling the jobs $\mathcal{J} \setminus \mathcal{J}^+$ that arrived but were not predicted to arrive and the cost of the jobs $\hat{\mathcal{J}} \setminus \mathcal{J}^+$ that were predicted to arrive but did not arrive. This prediction error is upper bounded by the prediction error in [7] for the problem with uniform deadlines.

**Extension to jobs that are approximately predicted correctly.** We generalize our algorithm and the previous result to allow the correctly predicted jobs $\mathcal{J}^+$ to include jobs that are approximately predicted correctly, where the tolerable approximation is parameterized by a parameter chosen by the algorithm designer. The result for this extension requires an additional smoothness condition on the quality cost $F(S, \mathcal{J})$ of a schedule. This condition is satisfied for the flow time problem, but not by the one with deadlines.[1]

**Experiments.** In Section 5, we show that when the prediction error is small, our algorithm empirically outperforms on both real and synthetic datasets the online algorithm that achieves the optimal competitive ratio for energy plus flow time minimization without predictions.

## 1.2 Related work

**Energy-efficient scheduling.** Energy-efficient scheduling was initiated by Yao et al. [28], who studied the energy minimization with deadlines problem in both offline and online settings. These offline and online algorithms were later improved in [13, 10]. Over the last two decades, energy-efficient scheduling has been extended to several other objective functions. In particular, Albers and Fujiwara [2] proposed the problem of energy plus flow time minimization, which has been studied extensively (see, e.g., [3, 8, 14, 12, 18, 11, 9]).

**Learning-augmented algorithms.** Algorithms with predictions is a recent and extremely active area, especially in online algorithms, where it was initiated in [22, 27]. Many different scheduling problems have been studied with predictions (see, e.g., [19, 24, 17, 20, 6, 5, 16, 21]).

**Learning-augmented energy-efficient scheduling.** Energy-efficient scheduling with predictions has been studied by Antoniadis et al. [4] and Bamas et al. [7], who focus on the problem with deadlines, which is a special case of our setting. The prediction model in Bamas et al. [7] is the closest to ours. For the problem with deadlines, the algorithm in [7] achieves a better consistency-robustness tradeoff than our algorithm, but their algorithm and prediction model do not extend to more general energy-efficient scheduling problems such as the flow time problem. In addition, the competitive ratio as a function of the prediction error is only obtained in [7] in the case of uniform deadlines where the difference between the deadline and release date of a job is equal for all jobs (the authors mention that defining algorithms for general deadlines becomes complex and notationally heavy when aiming for bounds as a function of the prediction error). Thanks to our algorithmic framework and definition of prediction error, our bound generalizes to the non-uniform deadlines without complicating our algorithm. Antoniadis et al. [4] propose a significantly different prediction model that requires an equal number of jobs in both the prediction $\hat{\mathcal{J}}$ and true set of jobs $\mathcal{J}$. Consequently, their results are incomparable to ours and those by Bamas et al. [7].

---

[1]We note that Bamas et al. [7] give an alternate approach to transform an arbitrary algorithm with predictions for the problem with uniform deadlines to an algorithm that allows small deviations in the release time of the jobs. This approach can be applied to our algorithm for the problem with uniform deadlines.

## 2 Preliminaries

In the *General Energy-Efficient Scheduling (GES)* problem, an instance is described by a collection $\mathcal{J}$ of $n$ jobs and a real-valued cost function $F(S, \mathcal{J})$ that takes as input the instance $\mathcal{J}$ and a schedule $S$ for $\mathcal{J}$, and returns some quality evaluation of the schedule. Each job $(j, r_j, p_j) \in \mathcal{J}$ consists of a release time $r_j$, a processing time $p_j$, and an identifier $j$ (and potentially other parameters such as weights $v_j$ and deadlines $d_j$). We often abuse notation and write $j \in \mathcal{J}$ instead of $(j, r_j, p_j) \in \mathcal{J}$. For any time interval $I$, we let $\mathcal{J}_I = \{j \in \mathcal{J} : r_j \in I\}$ be the subset of jobs of $\mathcal{J}$ with release time in $I$. For intervals $I = [0, t]$ or $I = [t, \infty]$, we write $\mathcal{J}_{\leq t}$ and $\mathcal{J}_{\geq t}$.

A *feasible schedule* for a set of jobs $\mathcal{J}$ is specified by $S = \{s_j(t)\}_{t \geq 0, j \in \mathcal{J}_{\leq t}}$, where $s(t) := \sum_{j \in \mathcal{J}_{\leq t}} s_j(t)$ is the speed at which the machine runs at time $t$. Thus, $s_j(t)/s(t)$ is the fraction of the processing power of the machine allocated to job $j$ at time $t$.[2] During a time interval $I$, there are $\int_I s_j(t)\mathrm{dt}$ units of work for job $j$ that are completed and we let $S_I$ be the sub-schedule $\{s_j(t)\}_{t \in I, j \in \mathcal{J}_I}$. The cost function we consider is a combination of energy consumption and quality cost for the output schedule. The energy consumption incurred by a schedule is $E(S) = \int_{t \geq 0} s(t)^\alpha \mathrm{dt}$, where $\alpha > 1$ is a problem-dependent constant, chosen so that the power at time $t$ is $s(t)^\alpha$. To define the quality of a schedule, we introduce the *work profile* $W_j^S := \{w_j^S(t)\}_{t \geq r_j}$ of schedule $S$ for job $j$, where $w_j^S(t) := p_j - \int_{r_j}^t s_j(u)\mathrm{du}$ is the amount of work for $j$ remaining at time $t$.

We consider general objective functions of the form $\mathrm{cost}(S, \mathcal{J}) = E(S) + F(S, \mathcal{J})$ and the goal is to compute a feasible schedule of minimum cost. $F(S, \mathcal{J}) = f((W_1^S, j_1), \ldots, (W_n^S, j_n))$ is an arbitrary *quality cost* function that is a function of the work profiles and the jobs' parameters. In the energy minimization with deadlines problem, $F(S, \mathcal{J}) = \infty$ if there is a job $j$ with completion time $c_j^S$ such that $c_j^S > d_j$, and $F(S, \mathcal{J}) = 0$ otherwise. In the energy plus flow time minimization problem, we have $F(S, \mathcal{J}) = \sum_{j \in \mathcal{J}} c_j^S - r_j$ (see Section 3.3 for additional functions $F$). A function $F(S, \mathcal{J})$ is subadditive if for all sets of jobs $\mathcal{J}_1$ and $\mathcal{J}_2$, we have $F(S, \mathcal{J}_1 \cup \mathcal{J}_2) \leq F(S, \mathcal{J}_1) + F(S, \mathcal{J}_2)$. $F$ is monotone if for all sets of jobs $\mathcal{J}$ and schedules $S$ and $S'$ such that $w_j^S(t) \leq w_j^{S'}(t)$ for all $j \in \mathcal{J}$ and $t \geq r_j$, we have that $F(S, \mathcal{J}) \leq F(S', \mathcal{J})$. We assume throughout the paper that $F$ is monotone subadditive, which holds for the deadlines and flow time problems. We let $S^*(\mathcal{J})$ and $\mathrm{OPT}(\mathcal{J}) := \mathrm{cost}(S^*(\mathcal{J}), \mathcal{J})$ be an optimal offline schedule and the optimal objective value.

**The general energy-efficient scheduling with predictions problem.** We augment the GES problem with predictions regarding future job arrivals and call this problem the General Energy-Efficient Scheduling with Predictions problem (GESP). In this problem, the algorithm is given at time $t = 0$ a prediction $\hat{\mathcal{J}} = \{(j, \hat{r}_j, \hat{p}_j)\}$ regarding the jobs $\mathcal{J} = \{(j, r_j, p_j)\}$ that arrive online. An important feature of our prediction model is that the number of predicted jobs $|\hat{\mathcal{J}}|$ can differ from the number of true jobs $|\mathcal{J}|$.

Next, we define a measure for the prediction error which generalizes the prediction error in [7] for the problem with uniform deadlines to any GES problem. With $\mathcal{J}^+ = \mathcal{J} \cap \hat{\mathcal{J}}$ being the correctly predicted jobs, we define the prediction error as

$$\eta(\mathcal{J}, \hat{\mathcal{J}}) = \frac{1}{\mathrm{OPT}(\hat{\mathcal{J}})} \max\{\mathrm{OPT}(\mathcal{J} \setminus \mathcal{J}^+), \mathrm{OPT}(\hat{\mathcal{J}} \setminus \mathcal{J}^+)\},$$

where $\mathrm{OPT}(\mathcal{J} \setminus \mathcal{J}^+)$ is the optimal cost of scheduling the true jobs $(j, r_j, p_j)$ such that either the prediction for $j$ was wrong or there was no prediction for $j$ and that $\mathrm{OPT}(\hat{\mathcal{J}} \setminus \mathcal{J}^+)$ is the optimal cost of scheduling the predicted jobs $(j, \hat{r}_j, \hat{p}_j)$ such that either the prediction for $j$ was wrong or $j$ never arrived. The prediction error $\eta(\mathcal{J}, \hat{\mathcal{J}})$ is then the maximum of these costs, normalized by the optimal cost $\mathrm{OPT}(\hat{\mathcal{J}})$ of scheduling the predicted jobs. We assume that $\hat{\mathcal{J}} \neq \emptyset$ to ensure that $\eta(\mathcal{J}, \hat{\mathcal{J}})$ is well-defined. This prediction error is upper bounded by the prediction error $||w^{\mathrm{true}} - w^{\mathrm{pred}}||_\alpha^\alpha$ considered in [7] for the problem with uniform deadlines, which we prove in Appendix F.1. Here $w^{\mathrm{true}}$ and $w^{\mathrm{pred}}$ are the true and predicted workload at each time step $t$, i.e., the sum of the processing times of the jobs that arrive at $t$.

---

[2]For ease of notation, we allow the machine to split its processing power at every time step $t$ over multiple jobs. In practice, this is equivalent to partitioning time into arbitrarily small time periods and splitting each time period into smaller subperiods such that the machine is processing one job during each subperiod.

We note that in the above error model, a job $j$ is in the set of correctly predicted jobs $\mathcal{J}^+$ only if all the parameters of $j$ have been predicted exactly correctly. To overcome this limitation, we introduce in Section 4 a more general error model where some small deviations between the true and predicted parameters of a job $j$ are allowed for the correctly predicted jobs $\mathcal{J}^+$. In Appendix F.1, we provide further discussion of this prediction model in comparison with [7, 4].

**Performance metrics.** The standard evaluation metrics for an online algorithm with predictions are its consistency, robustness, and competitive ratio as a function of the prediction error [23, 22]. The competitive ratio of an algorithm ALG as a function of a prediction error $\eta$ is $c(\eta) = \max_{\mathcal{J}, \hat{\mathcal{J}}: \eta(\mathcal{J}, \hat{\mathcal{J}}) \leq \eta} \frac{\text{cost}_{\text{ALG}}(\mathcal{J}, \hat{\mathcal{J}})}{\text{OPT}(\mathcal{J})}$. ALG is $\rho$-robust if for all $\eta \geq 0$, $c(\eta) \leq \rho$ (competitive ratio when the error is arbitrarily large) and $\mu$-consistent if $c(0) \leq \mu$ (competitive ratio when the prediction is exactly correct). We say that the competitive ratio of ALG is smooth if it smoothly degrades from $\mu$ to $\rho$ as the prediction error $\eta$ grows.

# 3 The Algorithm

In this section, we develop a simple and general algorithmic framework for GESP and analyze the resulting consistency, robustness, and competitive ratio as a function of the prediction error. We first note that the algorithm with predictions from [7] for the problem with deadlines does not easily generalize to some of the other problems that we consider, including the flow time problem (see Appendix F.3 for additional discussion). A major difference is that our algorithm consists of two distinct phases.

**Predictions cannot be completely trusted.** We also note that a first natural approach is to assume that the predictions are exactly correct and aim for a 1-consistent algorithm. For the problem with deadlines, Bamas et al. [7] showed that there is no 1-consistent algorithm with bounded robustness. In Appendix A.1, we show that this approach would also fail for the flow time problem because the algorithm might start by processing jobs too fast and consume too much energy when trusting the predictions. More generally, in Appendix A.2, we show that there is a necessary trade-off between consistency and robustness for the flow time problem by proving that any $1 + \lambda$-consistent algorithm must be $O(\sqrt{1 + 1/\lambda})$-robust.

## 3.1 Description of the algorithm

The algorithm, called TPE, takes as input an arbitrary quality of service cost function $F$, predictions $\hat{\mathcal{J}}$, a confidence level $\lambda \in (0, 1]$ in the predictions, an offline algorithm OFFLINEALG for $F$, and an online algorithm ONLINEALG for $F$ (without predictions). We denote by $\text{OFF}(\mathcal{J}) := \text{cost}(\text{OFFLINEALG}(\mathcal{J}), \mathcal{J})$ the objective value achieved by OFFLINEALG over $\mathcal{J}$.

---

**Algorithm 1** Two-Phase Energy Efficient Scheduling (TPE)

---

**Input:** predicted and true sets of jobs $\hat{\mathcal{J}}$ and $\mathcal{J}$, quality of cost function $F$, offline and online algorithms (without predictions) OFFLINEALG and ONLINEALG for problem $F$, confidence level $\lambda \in (0, 1]$.

1: **for** $t \geq 0$ **do**
2:     **if** $\text{OFF}(\mathcal{J}_{\leq t}) > \lambda \cdot \text{OFF}(\hat{\mathcal{J}})$ **then**
3:         $t_\lambda \leftarrow t$
4:         **break**
5:     $\{s_j(t)\}_{j \in \mathcal{J}_{\leq t}} \leftarrow \text{ONLINEALG}(\mathcal{J}_{\leq t})(t)$
6: $\{\hat{s}_j(t)\}_{t \geq t_\lambda, j \in \hat{\mathcal{J}}_{\geq t_\lambda}} \leftarrow \text{OFFLINEALG}(\hat{\mathcal{J}}_{\geq t_\lambda})$
7: **for** $t \geq t_\lambda$ **do**
8:     $\{s_j(t)\}_{j \in \mathcal{J}_{\leq t} \setminus \hat{\mathcal{J}}_{\geq t_\lambda}} \leftarrow \text{ONLINEALG}(\mathcal{J}_{\leq t} \setminus \hat{\mathcal{J}}_{\geq t_\lambda})(t)$
9:     $\{s_j(t)\}_{j \in \mathcal{J}_{[t_\lambda, t]} \cap \hat{\mathcal{J}}_{\geq t_\lambda}} \leftarrow \{\hat{s}_j(t)\}_{j \in \mathcal{J}_{[t_\lambda, t]} \cap \hat{\mathcal{J}}_{\geq t_\lambda}}$
10: **return** $\{s_j(t)\}_{t \geq 0, j \in \mathcal{J}}$

---

The algorithm proceeds in two phases. In the first phase (Lines 1-5), TPE ignores the predictions and runs the auxiliary online algorithm ONLINEALG over the true jobs $\mathcal{J}_{\leq t}$ that have been released by time $t$. More precisely, during the first phase of the algorithm, $s_j(t)$ is the speed according to the online algorithm ONLINEALG for all jobs. The first phase ends at the time $t_\lambda$ when the cost of the offline schedule computed by running OFFLINEALG on jobs $\mathcal{J}_{\leq t}$ reaches the threshold value $\lambda \cdot \text{OFF}(\hat{\mathcal{J}})$. As we will detail in the analysis section, this first phase guarantees a bounded robustness since we ensure that the offline cost for the true jobs reaches some value before starting to trust the predictions (hence, TPE does not initially 'burn' too much energy compared to the optimal offline cost, unlike the example described in Appendix A.1).

In the second phase (Lines 6-9), TPE starts leveraging the predictions. More precisely, TPE needs to set the speeds for three different types of jobs: (1) the remaining jobs that were correctly predicted (i.e., $\mathcal{J}_{\geq t_\lambda} \cap \hat{\mathcal{J}}_{\geq t_\lambda}$) (2) the remaining jobs that were not predicted (i.e., $\mathcal{J}_{\geq t_\lambda} \setminus \hat{\mathcal{J}}_{\geq t_\lambda}$) (3) the jobs that were not correctly scheduled in the first phase and still have work remaining at the switch point $t_\lambda$ (which are a subset of $\mathcal{J}_{< t_\lambda}$). To schedule these jobs, TPE combines two different schedules. The first one is the offline schedule $\hat{S} := \text{OFFLINEALG}(\hat{\mathcal{J}}_{\geq t_\lambda})$ for the jobs $\hat{\mathcal{J}}_{\geq t_\lambda}$ that are predicted to arrive in the second phase. Each future job in the true set that was correctly predicted (i.e., $j \in \mathcal{J}_{[t_\lambda, t]} \cap \hat{\mathcal{J}}_{\geq t_\lambda}$ on Line 9) will then be scheduled by following $\hat{S}$. The second schedule is an online schedule for the set of jobs $\mathcal{J} \setminus \hat{\mathcal{J}}_{\geq t_\lambda} = \mathcal{J}_{< t_\lambda} \cup \mathcal{J}_{\geq t_\lambda} \setminus \hat{\mathcal{J}}_{\geq t_\lambda}$, which includes all jobs that have not been completed during the first phase ($\subseteq \mathcal{J}_{< t_\lambda}$) and the incorrectly predicted jobs that are released during the second phase ($\mathcal{J}_{[t_\lambda, t]} \setminus \hat{\mathcal{J}}_{\geq t_\lambda}$). This online schedule is constructed by running ONLINEALG on the set $\mathcal{J} \setminus \hat{\mathcal{J}}_{\geq t_\lambda}$ (Line 8). Note that the total speed of the machine at each time step is the sum of the speeds of these two online and offline schedules.

## 3.2 Analysis of the algorithm

We analyze the competitive ratio of TPE as a function of the prediction error $\eta$, from which the consistency and robustness bounds follow. Missing proofs are provided in Appendix B. We separately bound the cost of the algorithm due to jobs in $\mathcal{J}_{< t_\lambda}$, $\mathcal{J}_{\geq t_\lambda} \setminus \hat{\mathcal{J}}_{\geq t_\lambda}$ and $\mathcal{J}_{\geq t_\lambda} \cap \hat{\mathcal{J}}_{\geq t_\lambda}$. We do this by analyzing the costs of schedules $S^{on} := \text{ONLINEALG}(\mathcal{J} \setminus \hat{\mathcal{J}}_{\geq t_\lambda})$ and $\hat{S} := \text{OFFLINEALG}(\hat{\mathcal{J}}_{\geq t_\lambda})$. In the next lemma, we first analyze the cost of combining, i.e., summing, two arbitrary schedules.

**Lemma 3.1.** *Let $\mathcal{J}_1$ be a set of jobs and $S_1$ be a feasible schedule for $\mathcal{J}_1$, let $\mathcal{J}_2$ be a set of jobs and $S_2$ be a feasible schedule for $\mathcal{J}_2$. Consider the schedule $S := S_1 + S_2$ for $\mathcal{J}_1 \cup \mathcal{J}_2$ which, at each time $t$, runs the machine at total speed $s(t) = s_1(t) + s_2(t)$ and processes each job $j \in \mathcal{J}_1$ at speed $s_{1,j}(t)$ and each job $j \in \mathcal{J}_2$ at speed $s_{2,j}(t)$. Then, $\text{cost}(S, \mathcal{J}_1 \cup \mathcal{J}_2) \leq \left( \text{cost}(S_1, \mathcal{J}_1)^{\frac{1}{\alpha}} + \text{cost}(S_2, \mathcal{J}_2)^{\frac{1}{\alpha}} \right)^\alpha.$*

We next upper bound the cost of the schedule output by TPE as a function of the prediction error $\eta$, which we decompose into $\eta_1 = \frac{\text{OPT}(\mathcal{J} \setminus \hat{\mathcal{J}})}{\text{OPT}(\hat{\mathcal{J}})}$ and $\eta_2 = \frac{\text{OPT}(\hat{\mathcal{J}} \setminus \mathcal{J})}{\text{OPT}(\hat{\mathcal{J}})}$. The proof uses the previous lemma repeatedly, first to analyze the cost of the schedule $S^{on} := \text{ONLINEALG}(\mathcal{J} \setminus \hat{\mathcal{J}}_{\geq t_\lambda})$ for the set of jobs $\mathcal{J} \setminus \hat{\mathcal{J}}_{\geq t_\lambda} = (\mathcal{J}_{< t_\lambda}) \cup (\mathcal{J}_{\geq t_\lambda} \setminus \hat{\mathcal{J}}_{\geq t_\lambda})$, then to analyze the cost of the final schedule, which combines $S^{on}$ and $\hat{S} := \text{OFFLINEALG}(\hat{\mathcal{J}}_{\geq t_\lambda})$.

**Lemma 3.2.** *Assume that OFFLINEALG is $\gamma_{off}$-competitive and that ONLINEALG is $\gamma_{on}$-competitive. Then, for all $\lambda \in (0, 1]$, the schedule $S$ output by TPE run with confidence parameter $\lambda$ satisfies $\text{cost}(S, \mathcal{J}) \leq \text{OPT}(\hat{\mathcal{J}}) \left( \gamma_{off}^{\frac{1}{\alpha}} + \gamma_{on}^{\frac{1}{\alpha}} ((\lambda \gamma_{off})^{\frac{1}{\alpha}} + \eta_1^{\frac{1}{\alpha}}) \right)^\alpha.$*

*Proof.* We start by upper bounding $\text{cost}(S^{on}, \mathcal{J} \setminus \hat{\mathcal{J}}_{\geq t_\lambda})$. First, by the algorithm, we have that $\text{OFF}(\mathcal{J}_{< t_\lambda}) \leq \lambda \cdot \text{OFF}(\hat{\mathcal{J}})$. Since OFFLINEALG is $\gamma_{off}$-competitive, we get

$$\text{OPT}(\mathcal{J}_{< t_\lambda}) \leq \text{OFF}(\mathcal{J}_{< t_\lambda}) \leq \lambda \cdot \text{OFF}(\hat{\mathcal{J}}) \leq \lambda \gamma_{off} \cdot \text{OPT}(\hat{\mathcal{J}}).$$

We also have that $\text{OPT}(\mathcal{J}_{\geq t_\lambda} \setminus \hat{\mathcal{J}}_{\geq t_\lambda}) \leq \text{OPT}(\mathcal{J} \setminus \hat{\mathcal{J}}) \leq \eta_1 \text{OPT}(\hat{\mathcal{J}})$ where the first inequality is since $\mathcal{J}_{\geq t_\lambda} \setminus \hat{\mathcal{J}}_{\geq t_\lambda} \subseteq \mathcal{J} \setminus \hat{\mathcal{J}}$ and the second is by definition of $\eta_1$. Recall that $S^*(.)$ denotes the optimal

offline schedule for the problem and consider the schedule $S' = S^\star(\mathcal{J}_{<t_\lambda}) + S^\star(\mathcal{J}_{\geq t_\lambda} \setminus \hat{\mathcal{J}}_{\geq t_\lambda})$ for $\mathcal{J} \setminus \hat{\mathcal{J}}_{\geq t_\lambda} = \mathcal{J}_{<t_\lambda} \cup (\mathcal{J}_{\geq t_\lambda} \setminus \hat{\mathcal{J}}_{\geq t_\lambda})$. We obtain that

$$\texttt{OPT}(\mathcal{J} \setminus \hat{\mathcal{J}}_{\geq t_\lambda}) \leq \text{cost}(S', \mathcal{J} \setminus \hat{\mathcal{J}}_{\geq t_\lambda}) \leq \left( \texttt{OPT}(\mathcal{J}_{<t_\lambda})^{\frac{1}{\alpha}} + \texttt{OPT}(\mathcal{J}_{\geq t_\lambda} \setminus \hat{\mathcal{J}}_{\geq t_\lambda})^{\frac{1}{\alpha}} \right)^\alpha$$

$$\leq \left( (\lambda \gamma_{\text{off}} \texttt{OPT}(\hat{\mathcal{J}}))^{\frac{1}{\alpha}} + (\eta_1 \texttt{OPT}(\hat{\mathcal{J}}))^{\frac{1}{\alpha}} ) \right)^\alpha$$

$$= \texttt{OPT}(\hat{\mathcal{J}}) \left( (\lambda \gamma_{\text{off}})^{\frac{1}{\alpha}} + \eta_1^{\frac{1}{\alpha}} \right)^\alpha,$$

where the second inequality is by Lemma 3.1. Since we assumed that ONLINEALG is $\gamma_{\text{on}}$-competitive,

$$\text{cost}(S^{on}, \mathcal{J} \setminus \hat{\mathcal{J}}_{\geq t_\lambda}) \leq \gamma_{\text{on}} \cdot \texttt{OPT}(\mathcal{J} \setminus \hat{\mathcal{J}}_{\geq t_\lambda}) \leq \gamma_{\text{on}} \cdot \texttt{OPT}(\hat{\mathcal{J}}) \left( (\lambda \gamma_{\text{off}})^{\frac{1}{\alpha}} + \eta_1^{\frac{1}{\alpha}} \right)^\alpha.$$

We now bound the cost of schedule $S$. First, note that $\text{cost}(\hat{S}, \hat{\mathcal{J}}_{\geq t_\lambda}) = \text{OFFLINEALG}(\hat{\mathcal{J}}_{\geq t_\lambda}) \leq \gamma_{\text{off}} \cdot \texttt{OPT}(\hat{\mathcal{J}}_{\geq t_\lambda}) \leq \gamma_{\text{off}} \cdot \texttt{OPT}(\hat{\mathcal{J}})$, where the first inequality is since OFFLINEALG is $\gamma_{\text{off}}$-competitive and the last one since $\hat{\mathcal{J}}_{\geq t_\lambda} \subseteq \hat{\mathcal{J}}$. Therefore, by applying again Lemma 3.1, we get:

$$\text{cost}(S, \mathcal{J}) \leq \left( \text{cost}(\hat{S}, \hat{\mathcal{J}}_{\geq t_\lambda})^{\frac{1}{\alpha}} + \text{cost}(S^{on}, \mathcal{J} \setminus \hat{\mathcal{J}}_{\geq t_\lambda})^{\frac{1}{\alpha}} \right)^\alpha$$

$$\leq \left( (\gamma_{\text{off}} \cdot \texttt{OPT}(\hat{\mathcal{J}}))^{\frac{1}{\alpha}} + \left( \gamma_{\text{on}} \cdot \texttt{OPT}(\hat{\mathcal{J}}) \left( (\lambda \gamma_{\text{off}})^{\frac{1}{\alpha}} + \eta_1^{\frac{1}{\alpha}} \right)^\alpha \right)^{\frac{1}{\alpha}} \right)^\alpha$$

$$= \texttt{OPT}(\hat{\mathcal{J}}) \left( \gamma_{\text{off}}^{\frac{1}{\alpha}} + \gamma_{\text{on}}^{\frac{1}{\alpha}} ((\lambda \gamma_{\text{off}})^{\frac{1}{\alpha}} + \eta_1^{\frac{1}{\alpha}}) \right)^\alpha. \qquad \square$$

We next state a simple corollary of Lemma 3.1.

**Corollary 3.3.** $\texttt{OPT}(\mathcal{J} \cap \hat{\mathcal{J}}) \geq \left( 1 - \eta_2^{\frac{1}{\alpha}} \right)^\alpha \texttt{OPT}(\hat{\mathcal{J}})$, and, assuming that OFFLINEALG is $\gamma_{\text{off}}$-competitive, we have: if $\texttt{OFF}(\mathcal{J}) \leq \lambda \texttt{OFF}(\hat{\mathcal{J}})$, then $\eta_2 \geq \left( 1 - (\lambda \gamma_{\text{off}})^{\frac{1}{\alpha}} \right)^\alpha$.

We are ready to state the main result of this section, which is our upper bound on the competitive ratio of TPE.

**Theorem 3.4.** *For any $\lambda \in (0, 1]$, TPE with a $\gamma_{on}$-competitive algorithm ONLINEALG and a $\gamma_{off}$-competitive offline algorithm OFFLINEALG achieves a competitive ratio of*

$$\begin{cases} \gamma_{on} & \text{if } \texttt{OFF}(\mathcal{J}) \leq \lambda \texttt{OFF}(\hat{\mathcal{J}}) \\ \dfrac{\left( \gamma_{off}^{\frac{1}{\alpha}} + \gamma_{on}^{\frac{1}{\alpha}} ((\lambda \gamma_{off})^{\frac{1}{\alpha}} + \eta_1^{\frac{1}{\alpha}}) \right)^\alpha}{\max\left\{ \frac{\lambda}{\gamma_{off}}, \eta_1 + \left( 1 - \eta_2^{\frac{1}{\alpha}} \right)^\alpha \right\}} & \text{otherwise.} \end{cases}$$

The consistency and robustness immediately follow (for simplicity, we present the results in the case where OFFLINEALG is optimal). Additional discussion on this competitive ratio is provided in Appendix 3.4.

**Corollary 3.5.** *For any $\lambda \in (0, 1)$, TPE with a $\gamma_{on}$-competitive algorithm ONLINEALG and an optimal offline algorithm OFFLINEALG is $1 + \gamma_{on} 2^\alpha \lambda^{\frac{1}{\alpha}}$ competitive if $\eta_1 = \eta_2 = 0$ (consistency) and $\max\{\gamma_{on}, \frac{1 + \gamma_{on} 2^\alpha \lambda^{\frac{1}{\alpha}}}{\lambda}\}$-competitive for all $\eta_1, \eta_2$ (robustness). In particular, for any constant $\epsilon > 0$, with $\lambda = (\frac{\epsilon}{\gamma_{on} 2^\alpha})^\alpha$, TPE is $1 + \epsilon$-consistent and $O(1)$-robust.*

### 3.3 Results for well-studied GES problems

We apply the general framework detailed in Section 3 to derive smooth, consistent and robust algorithms for a few classically studied objective functions.

**Energy plus flow time minimization.** Recall that $c_S^j$ denote the completion time of job $j$. The quality cost function is defined as: $F(S, \mathcal{J}) = \sum_{j \in \mathcal{J}} (c_S^j - r_j)$, with total objective $\text{cost}(S, \mathcal{J}) = F(S, \mathcal{J}) + E(S)$. By a direct application of Corollary 3.5, we get that for all $\lambda \in (0, 1]$, Algorithm 1

run with the 2-competitive online algorithm from [3] and confidence parameter $\lambda$ is $(1 + 2^{\frac{1}{\alpha}} \lambda^{\frac{1}{\alpha}})^\alpha$-consistent and $\frac{1 + 2 \cdot 2^{2\alpha} \lambda^{\frac{1}{\alpha}}}{\lambda}$-robust.

**Energy plus fractional weighted flow time minimization.** In this setting, each job has a weight $v_j$. The quality cost is $F(S, \mathcal{J}) = \sum_{j \in \mathcal{J}} v_j \int_{t \geq r_j} w_j^S(t) \mathrm{d}t$. We can use as ONLINEALG the 2-competitive algorithm from [14].

**Energy plus integral weighted flow time minimization.** In this setting, each job has a weight $v_j$. The quality cost function is defined as: $F(S, \mathcal{J}) = \sum_{j \in \mathcal{J}} v_j (c_S^j - r_j)$. We can use as ONLINEALG the $O((\alpha/\log \alpha)^2)$-competitive algorithm from [12].

**Energy minimization with deadlines.** In this setting, there is also a deadline $d_j$ for the completion of each job. By writing the quality cost as $F(S, \mathcal{J}) = \sum_{j \in \mathcal{J}} \delta_{c_S^j > d_j}$, where $\delta_{c_S^j > d_j} = +\infty$ if $c_S^j > d_j$ and 0 otherwise, the total objective can be written as $\mathrm{cost}(S, \mathcal{J}) = E(S) + F(S, \mathcal{J})$. We can use as ONLINEALG the AVERAGE RATE heuristic [28] (which is $2^\alpha$-competitive for uniform deadlines [7]). In particular, for uniform deadlines, and for all $\epsilon \in (0, 1]$, by setting $\lambda = (\frac{\epsilon}{C2^\alpha})^\alpha$, we obtain a consistency of $(1 + \epsilon)$ for a robustness factor of $O(4^{\alpha^2}/\epsilon^{\alpha-1})$.

### 3.4 Discussion on the competitive ratio

We assume in this section that OFFLINEALG is optimal. Note that for small $\eta_1$ and $\eta_2$, the competitive ratio is upper bounded as $\frac{\left(1 + \gamma_{\mathrm{on}}^{\frac{1}{\alpha}} (\lambda^{\frac{1}{\alpha}} + \eta_1^{\frac{1}{\alpha}})\right)^\alpha}{\eta_1 + \left(1 - \eta_2^{\frac{1}{\alpha}}\right)^\alpha}$, which smoothly goes to $(1 + \gamma_{\mathrm{on}}^{\frac{1}{\alpha}} \lambda^{\frac{1}{\alpha}})^\alpha$ (consistency case) when $\eta_1, \eta_2$ go to 0. Moreover, our upper bound distinguishes the effect of two possible sources of errors on the algorithm: (1) when removing jobs from the prediction ($\eta_1 = 0$ and $\eta_2$ goes to 1), the upper bound degrades monotonically to $O(\frac{1}{\lambda})$. (2) when adding jobs to the prediction ($\eta_2 = 0$ and $\eta_1$ goes to $+\infty$), the upper bound first degrades, then improves again, with an optimal asymptotic rate of $\gamma_{\mathrm{on}}$. This is since our algorithm mostly follows the online algorithm when the cost of the additional jobs dominates.

## 4 The Extension to Small Deviations

Note that in the definition of the prediction error $\eta$, a job $j$ is considered to be correctly predicted only if $r_j = \hat{r}_j$ and $p_j = \hat{p}_j$. In this extension, we consider that a job is correctly predicted even if its release time and processing time are shifted by a small amount. We also allow each job to have some weight $v_j > 0$, that can be shifted as well. Assuming an additional smoothness condition on the quality cost function $F(., .)$, which is satisfied for the energy plus flow time minimization problem and its variants, we propose and analyze an algorithm that generalizes the algorithm from the previous section.

The algorithm, called TPE-S and formally described in Appendix C, takes the same input parameters as Algorithm TPE, with some additional shift tolerance parameter $\eta^{\mathrm{shift}} \in [0, 1)$ that is chosen by the algorithm designer. Two main ideas are to artificially increase the predicted processing time $\hat{p}_j$ of each job $j$ (because the true processing time $p_j$ of job $j$ could be shifted and be slightly larger than $\hat{p}_j$) and to introduce small delays for the job speeds (because the true release time $r_j$ of some jobs j could be shifted and be slightly later than $\hat{r}_j$). Details can be found in Appendix C.

## 5 Experiments

We empirically evaluate the performance of Algorithm TPE-S on both synthetic and real datasets. Specifically, we consider the energy plus flow time minimization problem where $F(S, \mathcal{J}) = \sum_{j \in \mathcal{J}} c_j - r_j$ and consider unit-work jobs (i.e., $p_j = 1$ for all $j$) and fix $\alpha = 3$.

### 5.1 Experiment settings

**Benchmarks.** TPE-S is Algorithm 2 with the default setting $\lambda = 0.02$, $\eta^{\mathrm{shift}} = 1$ and $\sigma = 0.4$, where $\sigma$ is a parameter that controls the level of prediction error, that we call the error parameter.

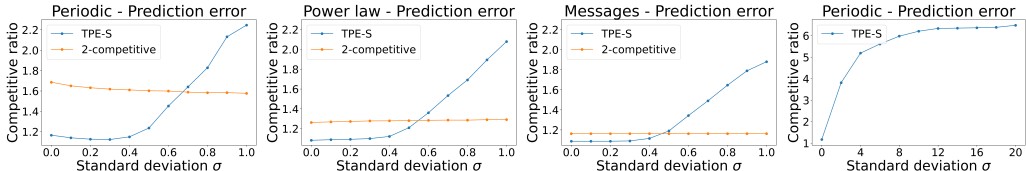

Figure 1: The competitive ratio achieved by our algorithm, TPE-S, and the benchmark algorithm, as a function of the error parameter $\sigma$ (from left-most to the second from the right), and the competitive ratio of TPE-S for a larger range of $\sigma$, as a function of $\sigma$ (right-most).

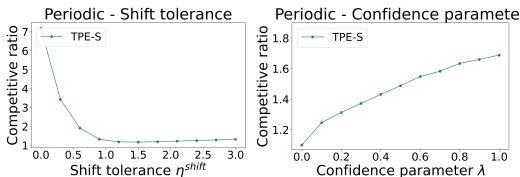

Figure 2: The competitive ratio achieved by our algorithm, TPE-S, as a function of the shift tolerance parameter $\eta^{\text{shift}}$ (left) and as a function of the confidence parameter $\lambda$ (right).

**2-COMPETITIVE** is the 2-competitive online algorithm from [3] that sets the speed at each time $t$ to $n(t)^{\frac{1}{\alpha}}$, where $n(t)$ is the number of jobs with $r_j \leq t$ unfinished at time $t$, and uses the Shortest Remaining Processing Time rule.

**Data sets.** We consider two synthetic datasets and a real dataset. For the synthetic data, we first generate a predicted set of jobs $\hat{\mathcal{J}}$ and we fix the value of the error parameter $\sigma > 0$. To create the true set of jobs $\mathcal{J}$, we generate, for each job $j \in \hat{\mathcal{J}}$, some error $err(j)$ sampled i.i.d. from $\mathcal{N}(0, \sigma)$. The true set of jobs is then defined as $\mathcal{J} = \{(j, \hat{r}_j + err(j)) : j \in \hat{\mathcal{J}}\}$, which is the set of all predicted jobs, shifted according to $\{err(j)\}$. Note that for all $j \in \mathcal{J}$, $j \in \mathcal{J}^{\text{shift}}$ only if $|\hat{r}_j - r_j| = |err(j)| < \frac{\eta^{\text{shift}}}{\beta(\hat{\mathcal{J}})} \cdot \frac{\texttt{OPT}(\hat{\mathcal{J}})}{|\hat{\mathcal{J}}|}$. Hence, a larger $\sigma > 0$ corresponds to a larger prediction error $\eta^g$. For the first synthetic dataset, called the periodic dataset, the prediction is a set of $n = 300$ jobs, with $i^{th}$ job's arrival $r_i = i/\alpha$. For the second synthetic dataset, we generate the prediction by using a power-law distribution. More precisely, for each time $t \in \{1, \ldots, T\}$, where we fix $T = 75$, the number of jobs' arrivals at time $t$ is set to $M(1 - p(a))$, where $p(a)$ is sampled from a power law distribution of parameter $a$, and $M$ is some scaling parameter. In all experiments, we use the values $a = 100$, $M = 500$.

We also evaluate the two algorithms on the College Message dataset from the SNAP database [26], where the scheduler must process messages that arrive over 9 days, each with between 300 and 500 messages. We first fix the error parameter $\sigma > 0$, then, for each day, we define the true set $\mathcal{J}$ as the arrivals for this day, and we create the predictions $\hat{\mathcal{J}}$ by adding some error $err(i)$ to the release time of each job $i$, where $err(i)$ is sampled i.i.d. from $\mathcal{N}(0, \sigma)$.

## 5.2 Experiment results

For each of the synthetic datasets, the competitive ratio achieved by the different algorithms is averaged over 10 instances generated i.i.d., and for the real dataset, it is averaged over the arrivals for each of the 9 days.

**Experiment set 1.** We first evaluate the performance of the algorithms as a function of the error parameter $\sigma$. In Figure 1, we observe that TPE-S outperforms 2-COMPETITIVE when the error parameter is small. In the right-most figure of Figure 1, the competitive ratio of TPE-S plateaus when the value of $\sigma$ increases, which is consistent with our bounded robustness guarantee.

**Experiment set 2.** In the second set of experiments, we study the impact of the parameters $\eta^{\text{shift}}$ and $\lambda$ of the algorithm for the periodic dataset (results for the other datasets can be found in Appendix E) and fix $\sigma = 0.4$. In the left plot of Figure 2, we observe the importance of allowing some shift in the predictions as the performance of our algorithms first rapidly improves as a function of $\eta^{\text{shift}}$

and then slowly deteriorates. The rapid improvement is because an increasing number of jobs are treated by the algorithm as being correctly predicted when $\eta^{\text{shift}}$ increases. Next, in the right plot, we observe that the competitive ratio deteriorates as a function of $\lambda$, which implies that the algorithm can completely skip the first phase that ignores the predictions and run the second phase that combines the offline and online schedules when the prediction error is not too large. Note, however, that a larger value of $\lambda$ leads to a better competitive ratio when the predictions are incorrect. Hence, there is a general trade-off here.

## 6    Limitations

The results in Section 3 and Section 4 require the quality cost function $F$ to be monotone subadditive, which holds for the flow time problem and the problem with deadlines but might not hold for some other energy-efficient scheduling problems. The results in Section 4 require an additional smoothness assumption on $F$, which holds for the flow time problem but not for the problem with deadlines. Finally, we have only tested our algorithm on the three datasets described in Section 5.

## Acknowledgements

Eric Balkanski was supported by NSF grants CCF-2210502 and IIS-2147361. Clifford Stein was supported in part by NSF grant CCF-2218677 and ONR grant ONR-13533312, and by the Wai T. Chang Chair in Industrial Engineering and Operations Research.

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
