# Appendix

## A  Consistent algorithms are not robust

In this section, we show that any learning-augmented algorithm for the (GSSP) problem must incur some trade-off between robustness and consistency. Note that some impossibility results for general objective functions of the form $\text{cost}(S, \mathcal{J}) = E(S) + F(S, \mathcal{J})$ given in Section 2 follow immediately from [7], since the problem of speed scaling with deadline constraints was studied there is a special case of (GSSP) (see Section 3.3).

We prove here some impossibility results for a different family of objective functions, where the objective is to maximize the total energy plus flow time. This is one of the most widely studied objectives of the form in $\text{cost}(S, \mathcal{J}) = E(S) + F(S, \mathcal{J})$ given in Section 2 (see for instance [2, 3, 12, 14]). Here, for all $j \in \mathcal{J}$, we let $c_S^j$ denote the completion time of job $j$ while following schedule $S$. The quality cost function studied in the remainder of this section is defined as: $F(S, \mathcal{J}) = \sum_{j \in \mathcal{J}} (c_S^j - r_j)$ and the total objective is $\text{cost}(S, \mathcal{J}) = F(S, \mathcal{J}) + E(S)$. We recall that for this problem, the best possible online algorithm is the 2-competitive algorithm from [3].

### A.1  Warm-up: no 1-consistent algorithm is robust

We show in this section that no algorithm that is perfectly consistent (i.e., achieves an optimal cost when the prediction is totally correct) can have a bounded competitive ratio in the case the prediction is incorrect. To show this property, we build an instance where a lot of jobs are predicted, but only one of them arrives. To achieve consistency, any online algorithm must 'burn' a lot of energy during the first few time steps; however, in the case where only one job arrives, the algorithm ends up having wasted too much energy. This illustrates the necessity of a trade-off between robustness and consistency.

**Proposition A.1.** *For the objective of minimizing total energy plus (non-weighted) flow time, there is no algorithm that is 1-consistent and $o(\sqrt{n})$-robust, even if all jobs have unit-size work and if $\mathcal{J} \subseteq \hat{\mathcal{J}}$.*

*Proof.* Set $\alpha = 2$ and consider an instance $(\hat{\mathcal{J}}, \mathcal{J}')$ where $\hat{\mathcal{J}}$ contains $n$ jobs of unit-size work such that the first job arrives at time $t = 0$ and the remaining $n - 1$ jobs arrive at time $t = \frac{1}{\sqrt{n}}$, and $\mathcal{J}'$ contains only the job that arrives at time $t = 0$.

By using results from [2], the optimal offline schedule for $\hat{\mathcal{J}}$ is to schedule each job $i \in [n]$ at speed $\sqrt{n - i + 1}$. Moreover, processing the first job any slower leads to a strictly worse cost. Hence, any algorithm that is 1-consistent (i.e, achieves an optimal competitive ratio when the realization is exactly $\hat{\mathcal{J}}$) must process the first job at speed $s_1(t) = \sqrt{n}$ for all $t \in [0, \frac{1}{\sqrt{n}}]$. In this case, the total objective is at least $\sqrt{n}^2 \cdot 1/\sqrt{n} + 1/\sqrt{n} = \sqrt{n} + 1/\sqrt{n}$.

However, by using results from [2], the optimal objective for $\mathcal{J}'$ is 2 (with the speed of the single job arriving at time $t = 0$ being set to 1). Hence, in the case where the realization is $\mathcal{J}'$, any algorithm that schedules the first job at speed $s_1(t) = \sqrt{n}$ has a competitive ratio at least $\frac{\sqrt{n} + 1/\sqrt{n}}{2}$.

Therefore, any 1-consistent algorithm must have a robustness factor of at least $\frac{\sqrt{n} + 1/\sqrt{n}}{2}$.   $\square$

### A.2  Consistency-robustness trade-off

In this section, we quantify more precisely the necessary trade-off between robustness and consistency. More precisely, we prove that there is a constant $C > 0$ such that for any $\lambda$ small enough, any algorithm that is at most $(1 + \lambda)$ consistent must be at least $C\sqrt{\frac{1}{\lambda} + 1}$ robust. Moreover, letting $n_{\leq t}(\mathcal{J}) = |\{j \in \mathcal{J} : r_j \leq t\}|$ be the number of jobs of $\mathcal{J}$ that arrived before time $t$, we show that for the following natural notion of error:

$$\tilde{\eta}(\mathcal{J}, \mathcal{J}') = \frac{1}{\max\{|\mathcal{J}|, |\mathcal{J}'|\}} \max_{t \geq 0}\{|n_{\leq t}(\mathcal{J}) - n_{\leq t}(\mathcal{J}')|\},$$

which mimics the probability density function of the predicted and realized jobs, this property remains true even if we assume a small prediction error $\tilde{\eta}(\mathcal{J}, \hat{\mathcal{J}})$. Hence, one cannot obtain a smooth algorithm relatively to this notion of error. This motivates the introduction of the more refined notion of error from Section 2. More specifically, we show the following lemma.

**Lemma A.2.** *For the objective of minimizing total energy plus (non-weighted) flow time, there are $\lambda' \in (0,1]$ and $C > 0$ such that for any $\epsilon > 0$, there is $M \in \mathbb{N}$ such that for all $\lambda \leq \lambda'$ and $n \geq M$, there is an instance $(\hat{\mathcal{J}}_{n,\lambda,\epsilon}, \mathcal{J}_{n,\lambda,\epsilon})$ such that $|\hat{\mathcal{J}}_{n,\lambda,\epsilon}| = n$ and $\tilde{\eta}(\hat{\mathcal{J}}_{n,\lambda,\epsilon}, \mathcal{J}_{n,\lambda,\epsilon}) \leq \epsilon$, and such that for any algorithm $\mathcal{A}$,*

- *either $cost_{\mathcal{A}}(\hat{\mathcal{J}} = \hat{\mathcal{J}}_{n,\lambda,\epsilon}, \mathcal{J} = \hat{\mathcal{J}}_{n,\lambda,\epsilon}) > (1+\lambda) \cdot OPT(\hat{\mathcal{J}}_{n,\lambda,\epsilon})$ (**large consistency factor**)*

- *or $cost_{\mathcal{A}}(\hat{\mathcal{J}} = \hat{\mathcal{J}}_{n,\lambda,\epsilon}, \mathcal{J} = \mathcal{J}_{n,\lambda,\epsilon}) \geq C\sqrt{\frac{1}{\lambda} + 1} \cdot OPT(\mathcal{J}_{n,\lambda,\epsilon})$ (**large robustness factor**).*

The rest of this section is dedicated to the proof of Lemma A.2.

We first describe our lower bound instance. In the remainder of this section, we set $\alpha = 2$.

**Lower bound instance.** Let $\lambda \in (0,1], n \in \mathbb{N}$ and $\epsilon > 0$. We construct an instance $(\hat{\mathcal{J}}_{n,\lambda,\epsilon}, \mathcal{J}_{n,\lambda,\epsilon})$ where the jobs in $\hat{\mathcal{J}}_{n,\lambda,\epsilon}$ can be organized in three different groups.

1. Group $A_{n,\lambda,\epsilon}$ is composed of $\frac{4}{3}\lambda\epsilon n$ jobs that all arrive at time 0.

2. Group $B_{n,\lambda,\epsilon}$ is composed of $\epsilon n$ jobs that all arrive at time $t_A := \frac{4\lambda\epsilon n}{3\sqrt{\epsilon n(1+\frac{4}{3}\lambda)}}$.

3. Group $C_{n,\lambda,\epsilon}$ consists of $n$ dummy jobs, where, for some $t' >> 0$, each job $j \in [n]$ arrives at time $t' + j$.

Next, we define $\mathcal{J}_{n,\lambda,\epsilon}$ as the union of jobs in $A_{n,\lambda,\epsilon}$ and $C_{n,\lambda,\epsilon}$. Note that by construction, we have $\tilde{\eta}(\hat{\mathcal{J}}_{n,\lambda,\epsilon}, \mathcal{J}_{n,\lambda,\epsilon}) \leq \epsilon$.

We now state and prove a few useful lemmas.

**Lemma A.3.** *Let $\mathcal{K}$ be a set of $n$ jobs that all arrive at some time $t \geq 0$. Then, we have*

$$\frac{4}{3}n^{3/2} \leq OPT(\mathcal{K}) \leq \frac{4}{3}n^{3/2} + o(n^{3/2}).$$

*Proof.* By [2], the optimal schedule is to run each job $i$ at speed $s_i = \sqrt{n-i+1}$. The total cost is as follows:

$$\begin{aligned}
cost(S^*(\mathcal{K})) &= F(S^*(\mathcal{K})) + E(S^*(\mathcal{K})) \\
&= \sum_{i=1}^{n}\sum_{j=1}^{i}\frac{1}{\sqrt{n-j+1}} + \sum_{i=1}^{n}\frac{1}{\sqrt{n-i+1}}\sqrt{n-i+1}^2 \\
&= \sum_{j=1}^{n}\frac{1}{\sqrt{n-j+1}}\sum_{i=j}^{n}1 + \sum_{i=1}^{n}\sqrt{n-i+1} \\
&= 2\sum_{i=1}^{n}\sqrt{n-i+1}
\end{aligned}$$

Hence we have

$$2\int_0^n \sqrt{x}dx \leq cost(S^*(\mathcal{K})) \leq 2\int_1^{n+1}\sqrt{x}dx$$

$$\Rightarrow \frac{4}{3}n^{3/2} \leq cost(S^*(\mathcal{K})) \leq \frac{4}{3}[(n+1)^{3/2}-1] = \frac{4}{3}[n^{3/2} + o(n^{3/2})]$$

$\square$

**Lemma A.4.** *Let $\mathcal{K} = \{j_1, \ldots, j_{|\mathcal{K}|}\}$ be a set of $|\mathcal{K}|$ jobs such that for all $i \in [|\mathcal{K}|]$, $|r_{i+1} - r_i| \geq 1$. Then, we have*

$$\mathit{OPT}(\mathcal{K}) = 2|\mathcal{K}|.$$

*Proof.* By using [2], the optimal solution is to run each job at speed 1. The result follows immediately.
□

**Lemma A.5.** *Let $\lambda \in (0, 1], n \in \mathbb{N}$. Then, the optimal cost for jobs in $\hat{\mathcal{J}}_{n,\lambda,\epsilon}$ is upper bounded as follows:*

$$\mathit{OPT}(\hat{\mathcal{J}}_{n,\lambda,\epsilon}) \leq \frac{4}{3}(\epsilon n)^{3/2}(1 + \lambda + o(\lambda) + o(1))$$

$$\text{as } \lambda \to 0 \text{ (independently of } \epsilon), \ n \to +\infty \text{ (for a fixed } \epsilon).$$

*Proof.* Consider the schedule $S$ which runs jobs in $A_{n,\lambda,\epsilon}$ at speed $\sqrt{\epsilon n(1 + \frac{4}{3}\lambda)}$, and jobs in $B_{n,\lambda,\epsilon}$ at the optimal speeds for $\epsilon n$ jobs arriving at the same time, and jobs in $C_{n,\lambda,\epsilon}$ at speed 1. Consider the cost of $S$ for all jobs in $A_{n,\lambda,\epsilon}$. Note that all the jobs in $A_{n,\lambda,\epsilon}$ are finished by time $t_A = \frac{4\lambda\epsilon n}{3\sqrt{\epsilon n(1 + \frac{4}{3}\lambda)}}$.
Hence, we have

$$\begin{aligned}
\text{cost}(S_{[0,t_A]}, A_{n,\lambda,\epsilon}) &\leq F(S_{[0,t_A]}, A_{n,\lambda,\epsilon}) + E(S_{[0,t_A]}) \\
&\leq t_A \frac{4}{3}\lambda\epsilon n + t_A \left(\sqrt{\epsilon n(1 + \frac{4}{3}\lambda)}\right)^2 \\
&\leq \frac{\frac{4}{3}\lambda\epsilon n \cdot \frac{4}{3}\lambda\epsilon n}{\sqrt{\epsilon n(1 + \frac{4}{3}\lambda)}} + \frac{4}{3}\lambda\epsilon n\sqrt{\epsilon n(1 + \frac{4}{3}\lambda)} \\
&= (\epsilon n)^{3/2}o(\lambda) + (\epsilon n)^{3/2}(\frac{4}{3}\lambda\sqrt{(1 + \frac{4}{3}\lambda)}) \\
&= (\epsilon n)^{3/2}[\frac{4}{3}\lambda + o(\lambda)].
\end{aligned}$$

Let $t_B$ be the time at which $S$ finishes all jobs in $B_{n,\lambda,\epsilon}$. Recall that all $n$ jobs in $B_{n,\lambda,\epsilon}$ arrive at time $t_A$. By Lemma A.3, we have

$$\text{cost}(S_{[t_A,t_B]}, B_{n,\lambda,\epsilon}) \leq \frac{4}{3}(\epsilon n)^{3/2} + o((\epsilon n)^{3/2}),$$

Since all jobs in $C_{n,\lambda,\epsilon}$ arrive at time $t' >> t_B$, we have

$$\text{cost}(S_{\geq t_B}, C_{n,\lambda,\epsilon}) = 2n = o((\epsilon n)^{3/2}).$$

Therefore, when $\lambda$ goes to 0 and $n$ goes to $+\infty$, the total cost of $S$ is upper bounded as follows:

$$\begin{aligned}
\text{cost}(S, \hat{\mathcal{J}}_{n,\lambda,\epsilon}) &= \text{cost}(S_{[0,t_A]}, A_{n,\lambda,\epsilon}) + \text{cost}(S_{[t_A,t_B]}, B_{n,\lambda,\epsilon}) + \text{cost}(S_{\geq t_B}, C_{n,\lambda,\epsilon}) \\
&\leq \frac{4}{3}(\epsilon n)^{3/2}[1 + \lambda + o(\lambda) + o(1)].
\end{aligned}$$

□

**Lemma A.6.** *There is $\lambda' \in (0, 1]$ such that for any $\epsilon > 0$, there is $M \in \mathbb{N}$ such that for all $\lambda \leq \lambda'$ and $n \geq M$, and for any schedule $S$ for $\hat{\mathcal{J}}_{n,\lambda,\epsilon}$ which has at least $\lambda\epsilon n$ units of jobs from $A_{n,\lambda,\epsilon}$ remaining at time $t_A$, we have*

$$\frac{\text{cost}(S, \hat{\mathcal{J}}_{n,\lambda,\epsilon})}{\mathit{OPT}(\hat{\mathcal{J}}_{n,\lambda,\epsilon})} > \left(1 + \frac{1}{4}\lambda\right).$$

*Proof.* Let $\lambda \in (0,1]$ and $\epsilon > 0$, and let $S$ be a schedule for $\hat{\mathcal{J}}_{n,\lambda,\epsilon}$ which has at least $\lambda\epsilon n$ units of jobs from $A_{n,\lambda,\epsilon}$ remaining at time $t_A$.

Note that the cost of $S$ for times $t \geq t_A$ is at least the cost of an optimal schedule for the remaining $\lambda\epsilon n$ units of jobs from $A_{n,\lambda,\epsilon}$ and the $\epsilon n$ units of job from $B_{n,\lambda,\epsilon}$. By Lemma A.3, we thus get that:

$$\text{cost}(S, \hat{\mathcal{J}}_{n,\lambda,\epsilon}) \geq \text{cost}(S, A_{n,\lambda,\epsilon} \cup B_{n,\lambda,\epsilon}) \geq \frac{4}{3}((1+\lambda)\epsilon n)^{3/2}.$$

Now, by Lemma A.5, we get that when $\lambda$ goes to $0$ (independently of $\epsilon$) and $n$ goes to $+\infty$,

$$\text{cost}(S^*(\hat{\mathcal{J}}_{n,\lambda,\epsilon})) \leq \frac{4}{3}(\epsilon n)^{3/2}(1 + \lambda + o(\lambda) + o(1)).$$

Hence,

$$\begin{aligned}
\frac{\text{cost}(S, \hat{\mathcal{J}}_{n,\lambda,\epsilon})}{\text{cost}(S^*(\hat{\mathcal{J}}_{n,\lambda,\epsilon}))} &\geq \frac{\frac{4}{3}((1+\lambda)\epsilon n)^{3/2}}{\frac{4}{3}(\epsilon n)^{3/2}(1 + \lambda + o(\lambda) + o(1))} \\
&= \left(1 + \frac{3}{2}\lambda + o(\lambda)\right) \cdot (1 - \lambda - o(\lambda) - o(1)) \\
&= 1 + \frac{1}{2}\lambda - o(\lambda) - o(1).
\end{aligned}$$

Hence, there is $\lambda' \in (0,1]$ and $M \in \mathbb{N}$ (note that $\lambda' \in (0,1]$ is independent of $\epsilon$ while $M$ depends on it) such that if $\lambda \leq \lambda'$ and $n \geq M$, then

$$\frac{\text{cost}(S, \hat{\mathcal{J}}_{n,\lambda,\epsilon})}{\text{cost}(S^*(\hat{\mathcal{J}}_{n,\lambda,\epsilon}))} > \left(1 + \frac{1}{4}\lambda\right).$$

$\square$

**Lemma A.7.** *Let* $\lambda \in (0,1], n \in \mathbb{N}$. *Assume that* $S$ *schedules at least* $\frac{1}{3}\lambda\epsilon n$ *units of jobs from* $A_{n,\lambda,\epsilon}$ *from time $0$ to $t_A$. Then, there is a constant $C > 0$ such that*

$$cost(S, \mathcal{J}_{n,\lambda,\epsilon}) \geq C(\epsilon n)^{3/2}\lambda\sqrt{1+\lambda}.$$

*Proof.* For convenience of exposition, assume that $S$ schedules exactly $\frac{1}{3}\lambda\epsilon n$ units of jobs from $A_{n,\lambda,\epsilon}$ from time $0$ to $t$ (note that if $S$ schedules more work from $A_{n,\lambda,\epsilon}$, then the cost can only be higher). By using [2], the optimal solution is to schedule each job $i$ at speed $s_i = \sqrt{\frac{\epsilon\lambda n}{3} - i + c + 1}$, where $c$ is the unique constant such that

$$\sum_{i=1}^{\frac{\epsilon\lambda n}{3}} \frac{1}{\sqrt{\frac{\epsilon\lambda n}{3} - i + c + 1}} = t_A.$$

To lower bound $c$, note that we then have

$$\begin{aligned}
t_A &\geq \sum_{i=\frac{1}{2}\frac{\epsilon\lambda n}{3}}^{\frac{\epsilon\lambda n}{3}} \frac{1}{\sqrt{\frac{\epsilon\lambda n}{3} - \frac{1}{2}\frac{\epsilon\lambda n}{3} + c + 1}} \\
&= \frac{\epsilon\lambda n}{6} \frac{1}{\sqrt{\frac{\epsilon\lambda n}{6} + c + 1}}.
\end{aligned}$$

By definition of $t_A$, we get

$$\begin{aligned}
\frac{4\lambda\epsilon n}{3\sqrt{\epsilon n(1 + \frac{4}{3}\lambda)}} &\geq \frac{\epsilon\lambda n}{6} \frac{1}{\sqrt{\frac{\epsilon\lambda n}{6} + c + 1}} \\
\iff (\frac{4}{3}6)^2(\frac{\epsilon\lambda n}{6} + c + 1) &\geq (1 + \frac{4}{3}\lambda)\epsilon n \\
\iff c &\geq c_2\epsilon n - c_3\epsilon\lambda n - 1. \qquad \text{with } c_2 = \frac{1}{64}, c_3 = -\frac{1}{48} + \frac{1}{6}
\end{aligned}$$

And the corresponding energy consumption is:

$$\sum_{i=1}^{\frac{\epsilon\lambda n}{3}} \sqrt{\frac{\epsilon\lambda n}{3} - i + c + 1}$$

$$\geq \sum_{i=1}^{\frac{\epsilon\lambda n}{6}} \sqrt{\frac{\epsilon\lambda n}{3} - \frac{\epsilon\lambda n}{6} + c_2\epsilon n - c_3\epsilon\lambda n}$$

$$\geq \frac{\epsilon\lambda n}{6} \sqrt{\frac{\epsilon\lambda n}{3} - \frac{\epsilon\lambda n}{6} + c_2\epsilon n + (\frac{1}{48} - \frac{1}{6})\epsilon\lambda n}$$

$$= \frac{\epsilon\lambda n}{6} \sqrt{c_2\epsilon n + \frac{1}{48}\epsilon\lambda n}$$

$$\geq C(\epsilon n)^{3/2}\lambda\sqrt{1 + \lambda}. \qquad\qquad \text{(for some constant } C > 0)$$

Therefore, any schedule $S$ that completes $\lambda\epsilon n$ jobs before time $t$ has cost lower bounded as:

$$\text{cost}(S, \mathcal{J}_{n,\lambda,\epsilon}) \geq C(\epsilon n)^{3/2}\lambda\sqrt{1 + \lambda}.$$

$\square$

We are now ready to present the proof of Lemma A.2.

**Proof of Lemma A.2.** By Lemma A.6, we have that there is a constant $\lambda' \in (0, 1]$ such that for all $\epsilon > 0$, there is $M \in \mathbb{N}$ such that for any algorithm $\mathcal{A}$ and $n \geq M$, and when running $\mathcal{A}$ with predictions $\hat{\mathcal{J}} = \hat{\mathcal{J}}_{n,\lambda,\epsilon}$ and realization $\mathcal{J} \in \{\hat{\mathcal{J}}_{n,\lambda,\epsilon}, \mathcal{J}_{n,\lambda,\epsilon}\}$, then either $\mathcal{A}$ schedules at least $\frac{1}{3}\lambda\epsilon n$ units of jobs of $A_{n,\lambda,\epsilon}$ before time $t$, or the schedule $S$ output by $\mathcal{A}$ satisfies:

$$\frac{\text{cost}(S, \hat{\mathcal{J}}_{n,\lambda,\epsilon})}{\text{cost}(S^*(\hat{\mathcal{J}}_{n,\lambda,\epsilon}))} > \left(1 + \frac{1}{4}\lambda\right).$$

Hence, if $\mathcal{A}$ achieves a consistency of at most $\left(1 + \frac{1}{4}\lambda\right)$, $\mathcal{A}$ must schedule at least $\frac{1}{3}\lambda\epsilon n$ units of jobs of $A_{n,\lambda,\epsilon}$ before time $t$. However, we then have, by Lemma A.7, that for some constant $C > 0$,

$$\text{cost}_{\mathcal{A}}(\hat{\mathcal{J}} = \hat{\mathcal{J}}_{n,\lambda,\epsilon}, \mathcal{J} = \mathcal{J}_{n,\lambda,\epsilon}) \geq C(\epsilon n)^{3/2}\lambda\sqrt{1 + \lambda}.$$

On the other hand, assuming that $\mathcal{J} = \mathcal{J}_{n,\lambda,\epsilon}$, we get by Lemma A.3 and Lemma A.4 that

$$\text{OPT}(\mathcal{J}_{n,\lambda,\epsilon}) \leq \frac{4}{3}(\lambda\epsilon n)^{3/2} + o((\epsilon n)^{3/2}) + 2n.$$

Hence, we get that for some constant $C'' > 0$ and $n$ large enough,

$$\frac{\text{cost}_{\mathcal{A}}(\hat{\mathcal{J}} = \hat{\mathcal{J}}_{n,\lambda,\epsilon}, \mathcal{J} = \mathcal{J}_{n,\lambda,\epsilon})}{\text{OPT}(\mathcal{J}_{n,\lambda,\epsilon})} \geq \frac{C(\epsilon n)^{3/2}\lambda\sqrt{1 + \lambda}}{\frac{4}{3}(\lambda\epsilon n)^{3/2} + o((\epsilon n)^{3/2}) + 2n} \geq C''\sqrt{\frac{1}{\lambda} + 1}.$$

$\square$

# B  Missing analysis from Section 3

**Lemma 3.1.** *Let $\mathcal{J}_1$ be a set of jobs and $S_1$ be a feasible schedule for $\mathcal{J}_1$, let $\mathcal{J}_2$ be a set of jobs and $S_2$ be a feasible schedule for $\mathcal{J}_2$. Consider the schedule $S := S_1 + S_2$ for $\mathcal{J}_1 \cup \mathcal{J}_2$ which, at each time $t$, runs the machine at total speed $s(t) = s_1(t) + s_2(t)$ and processes each job $j \in \mathcal{J}_1$ at speed $s_{1,j}(t)$ and each job $j \in \mathcal{J}_2$ at speed $s_{2,j}(t)$. Then, $\text{cost}(S, \mathcal{J}_1 \cup \mathcal{J}_2) \leq \left(\text{cost}(S_1, \mathcal{J}_1)^{\frac{1}{\alpha}} + \text{cost}(S_2, \mathcal{J}_2)^{\frac{1}{\alpha}}\right)^{\alpha}.$*

*Proof.* We first upper bound the quality cost $F(S, \mathcal{J}_1 \cup \mathcal{J}_2)$ of the proposed schedule $S$. In each infinitesimal time interval $[t, t + dt]$ and for all $j \in \mathcal{J}_1$, $S$ processes $s_{1,j}(t)dt$ units of work of job $j$, and for each $j \in \mathcal{J}_2$, $S$ processes $s_{2,j}(t)dt$ units of work of job $j$. Hence $S$ processes exactly the same amount of work for each job $j \in \mathcal{J}_1$ (resp. $j \in \mathcal{J}_2$) as $S_1$ (resp. $S_2$). We thus get that for all $t \geq 0$,

$$w_j^S(t) = w_j^{S_1}(t) \text{ for all } j \in \mathcal{J}_1 \qquad \text{and} \qquad w_j^S(t) = w_j^{S_2}(t) \text{ for all } j \in \mathcal{J}_2. \qquad (1)$$

Therefore,

$$
\begin{aligned}
F(S, \mathcal{J}_1 \cup \mathcal{J}_2) &\leq F(S, \mathcal{J}_1) + F(S, \mathcal{J}_2) && (F \text{ is sub-additive}) \\
&= f\left(\{(W_j^S, j)\}_{j \in \mathcal{J}_1}\right) + f\left(\{(W_j^S, j)\}_{j \in \mathcal{J}_2}\right) \\
&= f\left(\{(W_j^{S_1}, j)\}_{j \in \mathcal{J}_1}\right) + f\left(\{(W_j^{S_2}, j)\}_{j \in \mathcal{J}_2}\right) && (\text{by } (1)) \\
&= F(S_1, \mathcal{J}_1) + F(S_2, \mathcal{J}_2).
\end{aligned}
$$

Next, we upper bound the energy consumption $E(S)$ of the proposed schedule $S$.

$$
\begin{aligned}
E(S) &= \int (s_1(t) + s_2(t))^\alpha dt \\
&= \sum_{i=0}^{\alpha} \binom{\alpha}{i} \int (s_1(t)^\alpha)^{\frac{i}{\alpha}} (s_2(t)^\alpha)^{\frac{\alpha - i}{\alpha}} dt \\
&\leq \sum_{i=0}^{\alpha} \binom{\alpha}{i} \left(\int (s_1(t))^\alpha dt\right)^{\frac{i}{\alpha}} \left(\int (s_2(t))^\alpha dt\right)^{\frac{\alpha - i}{\alpha}} && (\text{Hölder's inequality}) \\
&= E(S_1) + E(S_2) + \sum_{i=1}^{\alpha - 1} \binom{\alpha}{i} E(S_1)^{\frac{i}{\alpha}} E(S_2)^{\frac{\alpha - i}{\alpha}} \\
&\leq E(S_1) + E(S_2) + \sum_{i=1}^{\alpha - 1} \binom{\alpha}{i} \text{cost}(S_1)^{\frac{i}{\alpha}} \text{cost}(S_2)^{\frac{\alpha - i}{\alpha}}.
\end{aligned}
$$

Therefore, the total cost of schedule $S$ can be upper bounded as follows:

$$
\begin{aligned}
\text{cost}(S, \mathcal{J}_1 \cup \mathcal{J}_2) &= F(S, \mathcal{J}_1 \cup \mathcal{J}_2) + E(S) \\
&\leq F(S_1, \mathcal{J}_1) + E(S_1) + F(S_2, \mathcal{J}_2) + E(S_2) + \sum_{i=1}^{\alpha - 1} \binom{\alpha}{i} \text{cost}(S_1)^{\frac{i}{\alpha}} \text{cost}(S_2)^{\frac{\alpha - i}{\alpha}} \\
&= \left(\text{cost}(S_1, \mathcal{J}_1)^{\frac{1}{\alpha}} + \text{cost}(S_2, \mathcal{J}_2)^{\frac{1}{\alpha}}\right)^\alpha.
\end{aligned}
$$

$\square$

**Corollary 3.3.** $\text{OPT}(\mathcal{J} \cap \hat{\mathcal{J}}) \geq \left(1 - \eta_2^{\frac{1}{\alpha}}\right)^\alpha \text{OPT}(\hat{\mathcal{J}})$, and, assuming that OFFLINEALG is $\gamma_{off}$-competitive, we have: if $\text{OFF}(\mathcal{J}) \leq \lambda \text{OFF}(\hat{\mathcal{J}})$, then $\eta_2 \geq \left(1 - (\lambda \gamma_{off})^{\frac{1}{\alpha}}\right)^\alpha$.

*Proof.* We prove the first part of the Corollary by contradiction. Assume that $\text{OPT}(\mathcal{J} \cap \hat{\mathcal{J}}) < \left(1 - \eta_2^{\frac{1}{\alpha}}\right)^\alpha \text{OPT}(\hat{\mathcal{J}})$. Next, by definition of the error $\eta_2$, we have $\text{OPT}(\hat{\mathcal{J}} \setminus \mathcal{J}) = \eta_2 \cdot \text{OPT}(\hat{\mathcal{J}})$. Hence, by Lemma 3.1, there exists a schedule $S$ for $(\mathcal{J} \cap \hat{\mathcal{J}}) \cup (\hat{\mathcal{J}} \setminus \mathcal{J}) = \hat{\mathcal{J}}$ such that

$$
\begin{aligned}
\text{cost}(S, \hat{\mathcal{J}}) &\leq \left(\text{OPT}(\hat{\mathcal{J}} \setminus \mathcal{J})^{\frac{1}{\alpha}} + \text{OPT}(\mathcal{J} \cap \hat{\mathcal{J}})^{\frac{1}{\alpha}}\right)^\alpha \\
&< \left((\eta_2 \text{OPT}(\hat{\mathcal{J}}))^{\frac{1}{\alpha}} + \left(\left(1 - \eta_2^{\frac{1}{\alpha}}\right)^\alpha \text{OPT}(\hat{\mathcal{J}})\right)^{\frac{1}{\alpha}}\right)^\alpha \\
&= \text{OPT}(\hat{\mathcal{J}}),
\end{aligned}
$$

which contradicts the definition of $\text{OPT}(\hat{\mathcal{J}})$ and ends the proof of the first result.

We now show the second part of the Corollary.

Assume that $\text{OFF}(\mathcal{J}) \leq \lambda\text{OFF}(\hat{\mathcal{J}})$. Then, since OFFLINEALG is $\gamma_{\text{off}}$-competitive, we have

$$\text{OPT}(\mathcal{J}) \leq \text{OFF}(\mathcal{J}) \leq \lambda\text{OFF}(\hat{\mathcal{J}}) \leq \lambda\gamma_{\text{off}}\text{OPT}(\hat{\mathcal{J}}).$$

In particular, $\text{OPT}(\mathcal{J} \cap \hat{\mathcal{J}}) \leq \lambda\gamma_{\text{off}}\text{OPT}(\hat{\mathcal{J}})$. Next, assume by contradiction, that $\eta_2 < \left(1 - (\lambda\gamma_{\text{off}})^{\frac{1}{\alpha}}\right)^{\alpha}$, which implies that $\text{OPT}(\hat{\mathcal{J}}\backslash\mathcal{J}) < \left(1 - (\lambda\gamma_{\text{off}})^{\frac{1}{\alpha}}\right)^{\alpha}\text{OPT}(\hat{\mathcal{J}})$. Then, by Lemma 3.1, there exists a schedule $S$ for $(\mathcal{J} \cap \hat{\mathcal{J}}) \cup (\hat{\mathcal{J}} \setminus \mathcal{J}) = \hat{\mathcal{J}}$ such that

$$\begin{aligned}
\text{cost}(S, \hat{\mathcal{J}}) &\leq \left(\text{OPT}(\hat{\mathcal{J}} \setminus \mathcal{J})^{\frac{1}{\alpha}} + \text{OPT}(\mathcal{J} \cap \hat{\mathcal{J}})^{\frac{1}{\alpha}}\right)^{\alpha} \\
&< \left(((\lambda\gamma_{\text{off}})\text{OPT}(\hat{\mathcal{J}}))^{\frac{1}{\alpha}} + \left(\left(1 - (\lambda\gamma_{\text{off}})^{\frac{1}{\alpha}}\right)^{\alpha}\text{OPT}(\hat{\mathcal{J}})\right)^{\frac{1}{\alpha}}\right)^{\alpha} \\
&= \text{OPT}(\hat{\mathcal{J}}),
\end{aligned}$$

which contradicts the definition of $\text{OPT}(\hat{\mathcal{J}})$. Hence, $\eta_2 \geq \left(1 - (\lambda\gamma_{\text{off}})^{\frac{1}{\alpha}}\right)^{\alpha}$. $\qquad\square$

**Theorem 3.4.** *For any $\lambda \in (0, 1]$,* TPE *with a $\gamma_{on}$-competitive algorithm* ONLINEALG *and a $\gamma_{off}$-competitive offline algorithm* OFFLINEALG *achieves a competitive ratio of*

$$\begin{cases}
\gamma_{on} & \text{if } \textit{OFF}(\mathcal{J}) \leq \lambda\textit{OFF}(\hat{\mathcal{J}}) \\
\dfrac{\left(\gamma_{off}^{\frac{1}{\alpha}} + \gamma_{on}^{\frac{1}{\alpha}}\left((\lambda\gamma_{off})^{\frac{1}{\alpha}} + \eta_1^{\frac{1}{\alpha}}\right)\right)^{\alpha}}{\max\left\{\frac{\lambda}{\gamma_{off}}, \eta_1 + \left(1 - \eta_2^{\frac{1}{\alpha}}\right)^{\alpha}\right\}} & \textit{otherwise.}
\end{cases}$$

*Proof.* First, assume that that for all $t \geq 0$, $\text{OFF}(\mathcal{J}_{\leq t}) \leq \lambda\text{OFF}(\hat{\mathcal{J}})$ (i.e., TPE never goes through lines 6-10). Then, the schedule $S$ returned by the algorithm is obtained by running the $\gamma_{\text{on}}$-competitive algorithm ONLINEALG on $\mathcal{J}$, hence

$$\text{cost}(S, \mathcal{J}) \leq \gamma_{\text{on}} \cdot \text{OPT}(\mathcal{J}). \tag{2}$$

Next, assume that there is $t_\lambda \geq 0$ such that $\text{OFF}(\mathcal{J}_{\leq t_\lambda}) > \lambda\text{OFF}(\hat{\mathcal{J}})$. Since OFFLINEALG is $\gamma_{\text{off}}$-competitive, we immediately get:

$$\text{OPT}(\mathcal{J}) \geq \text{OPT}(\mathcal{J}_{\leq t_\lambda}) \geq \frac{\text{OFF}(\mathcal{J}_{\leq t_\lambda})}{\gamma_{\text{off}}} > \frac{\lambda}{\gamma_{\text{off}}}\text{OFF}(\hat{\mathcal{J}}) \geq \frac{\lambda}{\gamma_{\text{off}}}\text{OPT}(\hat{\mathcal{J}}).$$

By Corollary 3.3 and by definition of the error $\eta_1$, we also get the following lower bound on the optimal schedule:

$$\text{OPT}(\mathcal{J}) \geq \text{OPT}(\mathcal{J} \setminus \hat{\mathcal{J}}) + \text{OPT}(\mathcal{J} \cap \hat{\mathcal{J}}) \geq \eta_1\text{OPT}(\hat{\mathcal{J}}) + \left(1 - \eta_2^{\frac{1}{\alpha}}\right)^{\alpha}\text{OPT}(\hat{\mathcal{J}}).$$

Therefore,

$$\text{OPT}(\mathcal{J}) \geq \max\left\{\frac{\lambda}{\gamma_{\text{off}}}, \eta_1 + \left(1 - \eta_2^{\frac{1}{\alpha}}\right)^{\alpha}\right\}\text{OPT}(\hat{\mathcal{J}}). \tag{3}$$

Now, by Lemma 3.2, the cost of the schedule S output by TPE is always upper bounded as follows:

$$\text{cost}(S, \mathcal{J}) \leq \text{OPT}(\hat{\mathcal{J}})\left(\gamma_{\text{off}}^{\frac{1}{\alpha}} + \gamma_{\text{on}}^{\frac{1}{\alpha}}\left((\lambda\gamma_{\text{off}})^{\frac{1}{\alpha}} + \eta_1^{\frac{1}{\alpha}}\right)\right)^{\alpha}. \tag{4}$$

Hence, we get the following upper bound on the competitive ratio of TPE:

$$\frac{\text{cost}(S, \mathcal{J})}{\text{OPT}(\mathcal{J})}$$

$$= \mathbf{1}_{\text{OFF}(\mathcal{J}) \leq \lambda \text{OFF}(\hat{\mathcal{J}})} \frac{\text{cost}(S, \mathcal{J})}{\text{OPT}(\mathcal{J})} + \mathbf{1}_{\text{OFF}(\mathcal{J}) > \lambda \text{OFF}(\hat{\mathcal{J}})} \frac{\text{cost}(S, \mathcal{J})}{\text{OPT}(\mathcal{J})}$$

$$\leq \mathbf{1}_{\text{OFF}(\mathcal{J}) \leq \lambda \text{OFF}(\hat{\mathcal{J}})} \gamma_{\text{on}} + \mathbf{1}_{\text{OFF}(\mathcal{J}) > \lambda \text{OFF}(\hat{\mathcal{J}})} \frac{\left( \gamma_{\text{off}}^{\frac{1}{\alpha}} + \gamma_{\text{on}}^{\frac{1}{\alpha}} \left( (\lambda \gamma_{\text{off}})^{\frac{1}{\alpha}} + \eta_1^{\frac{1}{\alpha}} \right) \right)^{\alpha}}{\max \left\{ \frac{\lambda}{\gamma_{\text{off}}}, \eta_1 + \left( 1 - \eta_2^{\frac{1}{\alpha}} \right)^{\alpha} \right\}}. \qquad \text{by (2),(3),(4)}$$

$\square$

**Corollary 3.5.** *For any $\lambda \in (0, 1)$, TPE with a $\gamma_{on}$-competitive algorithm* ONLINEALG *and an optimal offline algorithm* OFFLINEALG *is $1 + \gamma_{on} 2^{\alpha} \lambda^{\frac{1}{\alpha}}$ competitive if $\eta_1 = \eta_2 = 0$ (consistency) and $\max\{\gamma_{on}, \frac{1 + \gamma_{on} 2^{2\alpha} \lambda^{\frac{1}{\alpha}}}{\lambda}\}$-competitive for all $\eta_1, \eta_2$ (robustness). In particular, for any constant $\epsilon > 0$, with $\lambda = (\frac{\epsilon}{\gamma_{on} 2^{\alpha}})^{\alpha}$, TPE is $1 + \epsilon$-consistent and $O(1)$-robust.*

*Proof.* We start by the consistency. Since we assumed that OFFLINEALG is optimal, by Corollary 3.3, we have that for all $\lambda \in (0, 1)$, $\text{OFF}(\mathcal{J}) > \lambda \text{OFF}(\hat{\mathcal{J}})$, when $\eta_2 = 0$. The result follows by an immediate upper bound on the competitive ratio in this case.

We now show the robustness. First note that if $\text{OFF}(\mathcal{J}) \leq \lambda \text{OFF}(\hat{\mathcal{J}})$, then $\frac{\text{cost}(S,\mathcal{J})}{\text{OPT}(\mathcal{J})} \leq \gamma_{\text{on}} \leq \max\{\gamma_{\text{on}}, \frac{1 + \gamma_{\text{on}} 2^{2\alpha} \lambda^{\frac{1}{\alpha}}}{\lambda}\}$ for any $\lambda \in [0, 1]$. Now, assume that $\text{OFF}(\mathcal{J}) > \lambda \text{OFF}(\hat{\mathcal{J}})$. We then have

$$\frac{\text{cost}(S, \mathcal{J})}{\text{OPT}(\mathcal{J})} \leq \frac{\left( 1 + \gamma_{\text{on}}^{\frac{1}{\alpha}} \left( \lambda^{\frac{1}{\alpha}} + \eta_1^{\frac{1}{\alpha}} \right) \right)^{\alpha}}{\max \left\{ \lambda, \eta_1 + \left( 1 - \eta_2^{\frac{1}{\alpha}} \right)^{\alpha} \right\}} \leq \frac{\left( 1 + \gamma_{\text{on}}^{\frac{1}{\alpha}} \left( \lambda^{\frac{1}{\alpha}} + \eta_1^{\frac{1}{\alpha}} \right) \right)^{\alpha}}{\max\{\lambda, \eta_1\}}.$$

If $\eta_1 \leq \lambda$, we get:

$$\frac{\text{cost}(S, \mathcal{J})}{\text{OPT}(\mathcal{J})} \leq \frac{\left( 1 + \gamma_{\text{on}}^{\frac{1}{\alpha}} (2\lambda^{\frac{1}{\alpha}}) \right)^{\alpha}}{\lambda} \leq \frac{1 + \gamma_{\text{on}} 2^{2\alpha} \lambda^{\frac{1}{\alpha}}}{\lambda},$$

and if $\eta_1 \geq \lambda$, we get:

$$\frac{\text{cost}(S, \mathcal{J})}{\text{OPT}(\mathcal{J})} \leq \frac{\left( 1 + \gamma_{\text{on}}^{\frac{1}{\alpha}} (2\eta_1^{\frac{1}{\alpha}}) \right)^{\alpha}}{\eta_1} \leq \frac{1 + \gamma_{\text{on}} 2^{2\alpha} \max\{\eta_1, \eta_1^{\frac{1}{\alpha}}\}}{\eta_1}.$$

Since the above function reaches its maximum value over $[\lambda, +\infty[$ for $\eta_1 = \lambda$, this immediately yields the result. $\square$

## C  The Extension with Job Shifts (Full version)

Note that in the definition of the prediction error $\eta$, a job $j$ is considered to be correctly predicted only if $r_j = \hat{r}_j$ and $p_j = \hat{p}_j$. In this section, we consider an extension where a job is considered to be correctly predicted even if the release time and processing time are shifted by a small amount. In this extension, we also allow each job to have some weight $v_j > 0$, that can be shifted as well. We propose and analyze an algorithm that generalizes the algorithm from the previous section.

**Motivating example.** Consider the objective of minimizing energy plus flow time with $\alpha = 2$. Let $(\hat{\mathcal{J}}, \mathcal{J})$ be an instance where $\mathcal{J}$ has $n$ jobs with weight $w = 1.01$ and processing time $p = 0.99$, all released at time $r = 0.1$, and $\hat{\mathcal{J}}$ has $n$ jobs with weight $w = 1$ and processing time $p = 1$, all released at time $r = 0$. Since $\hat{\mathcal{J}} \setminus \mathcal{J} = \hat{\mathcal{J}}$, we have here that $\eta_1 = \text{OPT}(\hat{\mathcal{J}} \setminus \mathcal{J}) = \text{OPT}(\hat{\mathcal{J}}) = \Omega(n^{3/2})$ (by Lemma A.3), whereas it seems reasonable to say that $\hat{\mathcal{J}}$ was a 'good' prediction for instance $\mathcal{J}$, since it accurately represents the pattern of the jobs in $\mathcal{J}$.

In this section, we assume that the quality of cost function $F$ is such that the total cost function $E + F$ satisfies a smoothness condition, which we next define.

**Smooth objective function.** Let $\mathbb{J}$ denote the collection of all sets of jobs. We say that a function $\beta : \mathbb{J} \longrightarrow \mathbb{R}$ is smooth if for all $\mathcal{J} \in \mathbb{J}$, $\{r'_j\}_{j \in \mathcal{J}} \geq 0$ and $\{\eta_j\}_{j \in \mathcal{J}} \geq 0$, we have $\beta(\mathcal{J}') \leq (1 + \max_j \eta_j)\beta(\mathcal{J})$, where $\mathcal{J}' = \{(j, r'_j, p_j(1 + \eta_j), v_j(1 + \eta_j)\}$. $\beta$ is monotone if for all $\mathcal{J}'' \subseteq \mathcal{J}$, we have $\beta(\mathcal{J}'') \leq \beta(\mathcal{J})$.

We say that a cost function $\text{cost}(.,.)$ is $\beta$-smooth if there is a smooth monotone function $\beta(.) \geq 1$ such that for all $\eta, \eta' \in [0, 1]$, $\mathcal{J}_1, \mathcal{J}_2$ with $|\mathcal{J}_1| = |\mathcal{J}_2|$ and bijection $\pi : \mathcal{J}_1 \longrightarrow \mathcal{J}_2$, and for all $S_1$ and $S_2$ feasible for $\mathcal{J}_1$ and $\mathcal{J}_2$:

- **(smoothness of optimal cost).** If for all $j \in \mathcal{J}_1$, $|r_j - r_{\pi(j)}| \leq \eta'$, $p_j \leq p_{\pi(j)}(1 + \eta)$ and $v_j \leq v_{\pi(j)}(1 + \eta)$, then

$$\text{OPT}(\mathcal{J}_1) \leq (1 + \beta(\mathcal{J}_1)\eta)\text{OPT}(\mathcal{J}_2) + \beta(\mathcal{J}_1)|\mathcal{J}_1|\eta'.$$

- **(shifted work profile for dominated schedule.)** If for all $j \in \mathcal{J}$, $p_j \leq p_{\pi(j)}$, $v_j \leq v_{\pi(j)}$, $r_j \geq r_{\pi(j)} - \eta'$ and for all $t \geq r_{\pi(j)} + \eta'$, $w^j_{S_1}(t) \leq w^{\pi(j)}_{S_2}(t - \eta')$, where $w^j_{S_1}(t)$ (resp. $w^j_{S_2}(t)$) denotes the remaining amount of work for $j$ a time $t$ for $S_1$ (resp. $S_2$), then

$$F(S_1, \mathcal{J}_1) \leq F(S_2, \mathcal{J}_2) + \beta(\mathcal{J}_1)|\mathcal{J}_1|\eta'.$$

In other words, if $\mathcal{J}_1$ and $\mathcal{J}_2$ are close to each other, then the optimal costs for $\mathcal{J}_1$ and $\mathcal{J}_2$ are close, and if schedules $S_1$, $S_2$ induce similar but slightly shifted work profiles for $\mathcal{J}_1$ and $\mathcal{J}_2$, then the quality costs for $S_1$ and $S_2$ are close.

We show in Appendix D that for the classically studied energy plus weighted flow time minimization problem with $\alpha \geq 1$, the cost function is $\max(4 \max_j v_j, 2^\alpha - 1)$-smooth. Note that for energy minimization with deadlines, the objective introduced in Section 3.3 is not smooth for any bounded function $\beta(.)$, since a small shift in the work profiles can induce a large increase in the objective function (in the case we miss a job's hard deadline). However, [7] (Section F.2) shows that it is also possible to transform any prediction-augmented algorithm for the energy plus deadline problem into a shift-tolerant algorithm.

**Shift tolerance and error definition.** In this extension, we allow each job in the prediction to be perturbed by a small amount. Past this tolerance threshold, the perturbed job is treated as a distinct job. We assume here that when a job arrives, it is always possible to identify which job of the prediction (if any) it corresponds to. More specifically, for each job $j$, we write $(j, r_j, p_j, v_j)$ for the real values of the parameters associated with $j$ and $(j, \hat{r}_j, \hat{p}_j, \hat{v}_j)$ for their predicted values (with the convention that $(j, r_j, p_j, v_j) = \emptyset$ if the job didn't arrive and $(j, \hat{r}_j, \hat{p}_j, \hat{v}_j) = \emptyset$ if the job was not predicted).

Next, we let $\eta^{\text{shift}} \in [0, 1)$ be a shift tolerance parameter, that is initially set by the decision-maker, and we assume that the objective function is $\beta$-smooth for some smooth monotone function $\beta(.) \geq 1$. We now define the set of 'correctly predicted' jobs as $\mathcal{J}^{\text{shift}} = \{(j, r_j, p_j, v_j) : |r_j - \hat{r}_j| \leq \frac{\eta^{\text{shift}}}{\beta(\hat{\mathcal{J}})} \cdot \frac{\text{OPT}(\hat{\mathcal{J}})}{|\hat{\mathcal{J}}|}, |p_j - \hat{p}_j| \leq \frac{\eta^{\text{shift}}}{\beta(\hat{\mathcal{J}})} \cdot \hat{p}_j, |v_j - \hat{v}_j| \leq \frac{\eta^{\text{shift}}}{\beta(\hat{\mathcal{J}})} \cdot \hat{v}_j\}$, which is the set of jobs whose release time, weights and processing times have only been slightly shifted as compared to their predicted values. The amount of shift we tolerate depends on the smoothness function $\beta(.)$ of the objective function $F$ and on the predicted instance $\hat{\mathcal{J}}$. In addition, note that the allowed shift in release time is proportional to the average cost per job (the intuition here is that for most objective functions, the average cost per job is at least the average completion time per job). We underscore the fact that $\mathcal{J} \setminus \mathcal{J}^{\text{shift}}$ contains both the predicted jobs that have past the shift tolerance and additional jobs in the realization. Finally, we let $\hat{\mathcal{J}}^{\text{shift}} = \{(j, \hat{r}_j, \hat{p}_j, \hat{v}_j) : (j, r_j, p_j, v_j) \in \mathcal{J}^{\text{shift}}\}$. The error $\eta^g = \frac{1}{\text{OPT}(\hat{\mathcal{J}})} \cdot \max\{\text{OPT}(\mathcal{J} \setminus \mathcal{J}^{\text{shift}}), \text{OPT}(\hat{\mathcal{J}} \setminus \hat{\mathcal{J}}^{\text{shift}})\}$ is now defined as the optimal cost for both the additional and missing jobs (similarly as in the previous sections) and the jobs that have past the shift tolerance, normalized by the optimal cost for the prediction.

**Algorithm description.** The Algorithm, called TPE-S and formally described in Algorithm 2, takes the same input parameters as Algorithm TPE, with some additional shift tolerance parameter $\eta^{\text{shift}} \in [0, 1)$, from which we compute the maximum allowed shift in release time $\bar{\eta}$ (Line 8).

TPE-S globally follows the structure of TPE, with a few differences, that we now detail. First, we start by slightly increasing the predicted weight and processing time of each job to obtain the set of jobs $\hat{\mathcal{J}}^{\text{up}} := \{(j, \hat{r}_j, \hat{p}_j(1 + \frac{\eta^{\text{shift}}}{\beta(\hat{\mathcal{J}})}), \hat{v}_j(1 + \frac{\eta^{\text{shift}}}{\beta(\hat{\mathcal{J}})}))\}$ (Line 1). Note that by the first smoothness condition, the optimal schedule for $\hat{\mathcal{J}}^{\text{up}}$ has only a slightly higher cost than $\text{OPT}(\hat{\mathcal{J}})$.

Then, similarly as TPE, TPE-S starts with a first phase where it follows the online algorithm ONLINEALG until the time $t_\lambda$ where the optimal offline has reached some threshold value $\lambda \text{OPT}(\hat{\mathcal{J}}^{\text{up}})$ (Lines 3-7). In the second phase (Lines 9-13), it again combines two schedules, this time, for (1) the jobs in $\mathcal{J}_{\geq t_\lambda}^{\text{shift}}$ that are within the shift tolerance (2) the jobs in $\mathcal{J} \setminus \mathcal{J}_{\geq t_\lambda}^{\text{shift}}$, which include the remaining jobs from phase 1 and the non-predicted jobs (or jobs that have past the shift-tolerance) that are released after time $t_\lambda$. To schedule the jobs in $\mathcal{J}_{\geq t_\lambda}^{\text{shift}}$, we first compute an offline schedule $\hat{S}$ for $\hat{\mathcal{J}}^{\text{up}}$ (Line 9). One small difference with TPE is that we will delay the schedule $\hat{S}$ by $\bar{\eta}$ time steps backwards when we schedule jobs in $\mathcal{J}_{\geq t_\lambda}^{\text{shift}}$. More precisely, each job $(j, r_j, p_j, v_j) \in \mathcal{J}_{\geq t_\lambda}^{\text{shift}}$ is scheduled on the same way the job with the same identifier $j$ in $\hat{\mathcal{J}}^{\text{up}}$ is scheduled by $\hat{S}$ at time $t - \bar{\eta}$ (Line 12). The intuition here is that we need to wait a small delay of $\bar{\eta}$ in order to identify which jobs of the predictions indeed arrived. Finally, and similarly as TPE, the speeds for jobs in $\mathcal{J} \setminus \mathcal{J}_{\geq t_\lambda}^{\text{shift}}$ are set by running ONLINEALG on the set $\mathcal{J} \setminus \mathcal{J}_{\geq t_\lambda}^{\text{shift}}$ (Line 13).

---

**Algorithm 2** Two-Phase Energy Efficient Scheduling with Shift Tolerance (TPE-S)

---

**Input:** predicted and true sets of jobs $\hat{\mathcal{J}}$ and $\mathcal{J}$, quality of cost function $F$, offline and online algorithms (without predictions) OFFLINEALG and ONLINEALG for problem $F$, confidence level $\lambda \in (0, 1]$, shift tolerance $\eta^{\text{shift}} > 0$.

1: $\hat{\mathcal{J}}^{\text{up}} \leftarrow \{(j, \hat{r}_j, \hat{p}_j(1 + \frac{\eta^{\text{shift}}}{\beta(\hat{\mathcal{J}})}), \hat{v}_j(1 + \frac{\eta^{\text{shift}}}{\beta(\hat{\mathcal{J}})}))\}$
2: $\hat{\text{OPT}} \leftarrow \text{OPT}(\hat{\mathcal{J}}^{\text{up}})$
3: **for** $t \geq 0$ **do**
4:     **if** $\text{OPT}(\mathcal{J}_{\leq t}) > \lambda \cdot \hat{\text{OPT}}$ **then**
5:         $t_\lambda \leftarrow t$
6:         **break**
7:     $\{s_j(t)\}_{j \in \mathcal{J}_{\leq t}} \leftarrow \text{ONLINEALG}(\mathcal{J}_{\leq t})(t)$
8: $\bar{\eta} \leftarrow \frac{\eta^{\text{shift}}}{\beta(\hat{\mathcal{J}})} \cdot \frac{\text{OPT}(\hat{\mathcal{J}})}{|\hat{\mathcal{J}}|}$
9: $\{\hat{s}_j(t)\}_{t \geq 0, j \in \hat{\mathcal{J}}^{\text{up}}} \leftarrow \text{OFFLINEALG}(\hat{\mathcal{J}}^{\text{up}})$
10: **for** $t \geq t_\lambda$ **do**
11:     **for** $j : (j, r_j, p_j, v_j) \in \mathcal{J}_{\geq t_\lambda}^{\text{shift}}$ **do**
12:         $s_j(t) \leftarrow \hat{s}_j(t - \bar{\eta})$
13:     $\{s_j(t)\}_{j \in \mathcal{J}_{\leq t} \setminus \mathcal{J}_{\geq t_\lambda}^{\text{shift}}} \leftarrow \text{ONLINEALG}(\mathcal{J}_{\leq t} \setminus \mathcal{J}_{\geq t_\lambda}^{\text{shift}})(t)$
14: **return** $\{s_j(t)\}_{t \geq 0, j \in \mathcal{J}}$

---

**Analysis.** We now present the analysis of TPE-S. All missing proofs are provided in Appendix D. In the following lemma, we start by upper bounding the cost of the schedule output by TPE-S for the jobs that were released in the second phase and were correctly predicted (i.e., within the shift tolerance). The proof mainly exploits the two smoothness conditions of the cost function.

**Lemma C.1.** *Assume that $cost(., .)$ is $\beta$-smooth. Consider the schedule $S^{shift}$, which, for all $t \geq t_\lambda$ and $(j, r_j, p_j, v_j) \in \mathcal{J}_{\geq t_\lambda}^{shift}$, processes job $j$ at speed*

$$s_j(t) = \hat{s}_j \left( t - \frac{\eta^{shift}}{\beta(\hat{\mathcal{J}})} \cdot \frac{OPT(\hat{\mathcal{J}})}{|\hat{\mathcal{J}}|} \right).$$

*Then,*

$$cost(S^{shift}, \mathcal{J}_{\geq t_\lambda}^{shift}) \leq (1 + 2\eta^{shift}(1 + \eta^{shift})) OPT(\hat{\mathcal{J}}).$$

We now show some slightly modified version of Corollary 3.3. Similarly as before, we write $\eta_1 = \frac{\text{OPT}(\mathcal{J} \setminus \mathcal{J}^{\text{shift}})}{\text{OPT}(\hat{\mathcal{J}})}$ to denote the error corresponding to additional jobs in the prediction, and $\eta_2 = \frac{\text{OPT}(\hat{\mathcal{J}} \setminus \hat{\mathcal{J}}^{\text{shift}})}{\text{OPT}(\hat{\mathcal{J}})}$ for the error corresponding to missing jobs.

**Corollary C.2.** *Assume that* $cost(.,.)$ *is* $\beta$-smooth, then

$$\text{OPT}(\mathcal{J}^{\text{shift}}) \geq \left[ \left(1 - \eta_2^{\frac{1}{\alpha}}\right)^\alpha - \eta^{\text{shift}} \right] \Big/ (1 + \eta^{\text{shift}}) \, \text{OPT}(\hat{\mathcal{J}}).$$

**Corollary C.3.** *Assume that* $cost(.,.)$ *is* $\beta$-smooth. *If* $\text{OPT}(\mathcal{J}) \leq \lambda \text{OPT}(\hat{\mathcal{J}})$, *then* $\eta_2 \geq \left(1 - (\lambda(1 + \eta^{\text{shift}}) + \eta^{\text{shift}})^{\frac{1}{\alpha}}\right)^\alpha$.

We now state the main result of this section, which is our upper bound on the competitive ratio of the shift-tolerant Algorithm TPE-S.

**Theorem C.4.** *Assume that* $cost(.,.)$ *is* $\beta$-smooth. *Then, for any* $\lambda \in (0,1], \eta^{\text{shift}} \in [0,1)$, *the competitive ratio of* TPE-S *run with trust parameter* $\lambda$, *a* $\gamma$-competitive algorithm ONLINEALG, *an optimal offline algorithm* OFFLINEALG, *and shift tolerance* $\eta^{\text{shift}}$ *is at most*

$$\begin{cases} \gamma & \text{if } \text{OPT}(\mathcal{J}) \leq \lambda \text{OPT}(\hat{\mathcal{J}}) \\[2ex] \dfrac{\left((1 + 2\eta^{\text{shift}}(1 + \eta^{\text{shift}}))^{\frac{1}{\alpha}} + \gamma^{\frac{1}{\alpha}} \left(\lambda^{\frac{1}{\alpha}} + \eta_1^{\frac{1}{\alpha}}\right)\right)^\alpha}{\max\left\{\lambda, \eta_1 + \left(\left(1 - \eta_2^{\frac{1}{\alpha}}\right)^\alpha - \eta^{\text{shift}}\right)(1 + \eta^{\text{shift}})^{-1}\right\}} & \text{otherwise.} \end{cases}$$

In particular, we deduce the following consistency and robustness guarantees.

**Corollary C.5.** *(consistency) For any* $\lambda \in (0,1]$ *and* $\eta^{\text{shift}} \in [0,1)$, *if* $\eta_1 = \eta_2 = 0$ *(all jobs are within the shift tolerance and there is no extra or missing jobs), then the competitive ratio of* TPE-S *run with trust parameter* $\lambda$ *and shift tolerance parameter* $\eta^{\text{shift}}$ *is upper bounded by* $\min(\frac{1}{\lambda}, \frac{(1 + \eta^{\text{shift}})}{(1 - \eta^{\text{shift}})}) \cdot \left((1 + 2\eta^{\text{shift}}(1 + \eta^{\text{shift}}))^{\frac{1}{\alpha}} + \gamma^{\frac{1}{\alpha}} \lambda^{\frac{1}{\alpha}}\right)^\alpha \leq \frac{(1 + 2\eta^{\text{shift}}(1 + \eta^{\text{shift}}))^2}{(1 - \eta^{\text{shift}})} \cdot (1 + \gamma 2^\alpha \lambda^{\frac{1}{\alpha}}).$

**Corollary C.6.** *(robustness) For any* $\lambda \in (0,1]$ *and* $\eta^{\text{shift}} \in [0,1)$, *the competitive ratio of Algorithm 1 run with trust parameter* $\lambda$ *and shift tolerance parameter* $\eta^{\text{shift}}$ *is upper bounded by* $\frac{(1 + 2\eta^{\text{shift}}(1 + \eta^{\text{shift}}))(1 + \gamma 2^{2\alpha} \lambda^{\frac{1}{\alpha}})}{\lambda}.$

## D  Missing analysis from Appendix C

**Lemma D.1.** *For the objective of minimizing total integral weighted flow time plus energy with* $\alpha \geq 1$, *the cost function is* $\max(4 \cdot \max_j v_j, 2^\alpha - 1)$-*smooth.*

*Proof.* Let $\mathcal{J}_1, \mathcal{J}_2$ with $|\mathcal{J}_1| = |\mathcal{J}_2|$, a bijection $\pi : \mathcal{J}_1 \longrightarrow \mathcal{J}_2$ and $S_1$ and $S_2$ feasible for $\mathcal{J}_1$ and $\mathcal{J}_2$.

We start with the first smoothness condition. Assume that for some $\eta, \eta' \in [0,1]$ and for all $j \in \mathcal{J}_1$, $|r_j - r_{\pi(j)}| \leq \eta', p_j \leq p_{\pi(j)}(1 + \eta)$ and $v_j \leq v_{\pi(j)}(1 + \eta)$. Let $S^*$ be an optimal schedule for $\mathcal{J}_2$ and consider the schedule $S = \{s_j(t) := (1 + \eta) \cdot s^*_{\pi(j)}(t - \eta')\}_{j \in \mathcal{J}_1}$ for $\mathcal{J}_1$.

Note that for all $j \in \mathcal{J}_1$ and $t \geq 0$, $s^*_{\pi(j)}(t - \eta') > 0$ only if $t - \eta' > r_{\pi(j)}$. Since we assumed $|r_j - r_{\pi(j)}| \leq \eta'$, we get that $s_j(t) > 0$ only if $t \geq r_j$, hence $S$ is feasible for $\mathcal{J}_1$. Next, note that

$$E(S) = E(S^*)(1 + \eta)^\alpha \leq E(S^*)(1 + (2^\alpha - 1)\eta).$$

Recall that $c^S_j$ denotes the completion time of $j$ by $S$. Since $p_j \leq (1 + \eta)p_{\pi(j)}$, and since we assumed that $|r_j - r_{\pi(j)}| \leq \eta'$, we have, by definition of $S$, that for all $j$, $c^S_j \leq \eta' + c^{S^*}_{\pi(j)}$.

Hence,

$$
\begin{aligned}
F(S, \mathcal{J}_1) &= \sum_{j \in \mathcal{J}_1} v_j(c_j^S - r_j) \\
&\leq \sum_{j \in \mathcal{J}_1} v_{\pi(j)}(1 + \eta)(c_{\pi(j)}^{S^*} + \eta' - (r_{\pi(j)} - \eta')) \\
&= (1 + \eta)F(S^*, \mathcal{J}_2) + 2 \max_j v_j |\mathcal{J}_1|(1 + \eta)\eta' \\
&\leq (1 + \eta)F(S^*, \mathcal{J}_2) + 4 \max_j v_j |\mathcal{J}_1|\eta' \qquad\qquad \eta \in [0, 1] \\
&\leq (1 + (2^\alpha - 1)\eta)F(S^*, \mathcal{J}_2) + 4 \max_j v_j |\mathcal{J}_1|\eta' \qquad \alpha \geq 1.
\end{aligned}
$$

Therefore,

$$
\begin{aligned}
\mathtt{OPT}(\mathcal{J}_1) &\leq \mathrm{cost}(S, \mathcal{J}_1) \\
&= E(S) + F(S, \mathcal{J}_1) \\
&\leq (E(S^*) + F(S^*, \mathcal{J}_2)) \cdot (1 + (2^\alpha - 1)\eta) + 4 \max_j v_j |\mathcal{J}_1|\eta' \\
&= \mathrm{cost}(S^*, \mathcal{J}_2)(1 + (2^\alpha - 1)\eta) + 4 \max_j v_j |\mathcal{J}_1|\eta' \\
&= \mathtt{OPT}(\mathcal{J}_2)(1 + (2^\alpha - 1)\eta) + 4 \max_j v_j |\mathcal{J}_1|\eta'.
\end{aligned}
$$

We now show the second smoothness condition. Assume that for all $j \in \mathcal{J}_1$, $p_j \leq p_{\pi(j)}$, $v_j \leq v_{\pi(j)}$, $r_j \geq r_{\pi(j)} - \eta'$ and that for all $t \geq r_{\pi(j)} + \eta'$, $w_{S_1}^j(t) \leq w_{S_2}^{\pi(j)}(t - \eta')$. Then, in particular $w_{S_1}^j(c_{\pi(j)}^{S_2} + \eta') \leq w_{S_2}^{\pi(j)}(c_{\pi(j)}^{S_2}) = 0$, hence $c_j^{S_1} = \min\{t \geq r_j : w_{S_1}^j(t) = 0\} \leq c_{\pi(j)}^{S_2} + \eta'$. By a similar argument as above, we conclude that:

$$
F(S, \mathcal{J}_1) \leq F(S_2, \mathcal{J}_2) + 4 \max_j v_j |\mathcal{J}_1|\eta'.
$$

$\square$

**Lemma C.1.** *Assume that $\mathrm{cost}(.,.)$ is $\beta$-smooth. Consider the schedule $S^{shift}$, which, for all $t \geq t_\lambda$ and $(j, r_j, p_j, v_j) \in \mathcal{J}_{\geq t_\lambda}^{shift}$, processes job $j$ at speed*

$$
s_j(t) = \hat{s}_j \left( t - \frac{\eta^{shift}}{\beta(\hat{\mathcal{J}})} \cdot \frac{\mathit{OPT}(\hat{\mathcal{J}})}{|\hat{\mathcal{J}}|} \right).
$$

*Then,*
$$
\mathrm{cost}(S^{shift}, \mathcal{J}_{\geq t_\lambda}^{shift}) \leq (1 + 2\eta^{shift}(1 + \eta^{shift}))\mathit{OPT}(\hat{\mathcal{J}}).
$$

*Proof.* For simplifying the exposition, in the remainder of the proof, we write $\bar{\eta}$ instead of $\frac{\eta^{shift}}{\beta(\hat{\mathcal{J}})} \cdot \frac{\mathtt{OPT}(\hat{\mathcal{J}})}{|\hat{\mathcal{J}}|}$.

We first analyse the energy cost.

$$
E(S^{shift}) = \int_{t \geq t_\lambda} \left( \sum_{j \in \mathcal{J}_{\geq t_\lambda}^{shift}} \hat{s}_j(t - \bar{\eta}) \right)^\alpha \mathrm{dt} = \int_{t \geq t_\lambda - \bar{\eta}} \left( \sum_{j \in \mathcal{J}_{\geq t_\lambda}^{shift}} \hat{s}_j(t) \right)^\alpha \mathrm{dt}
$$

$$
\leq \int_t \left( \sum_{j \in \hat{\mathcal{J}}^{up}} \hat{s}_j(t) \right)^\alpha \mathrm{dt} = E(\hat{S}). \qquad (5)
$$

Next, we analyze the quality cost. Note that by definition of $S^{\text{shift}}$, we have that for all $j \in \mathcal{J}_{\geq t_\lambda}^{\text{shift}}$ and $t \geq \hat{r}_j + \bar{\eta}$, the same amount of work for $(j, r_j, p_j, v_j)$ has been processed by $S^{\text{shift}}$ at time $t$ as the amount of work for $(j, \hat{r}_j, \hat{p}_j(1 + \frac{\eta^{\text{shift}}}{\beta(\hat{\mathcal{J}})}), \hat{v}_j(1 + \frac{\eta^{\text{shift}}}{\beta(\hat{\mathcal{J}})}))$ processed by $\hat{S}$ at time $t - \bar{\eta}$. Hence, for all $t \geq \hat{r}_j + \bar{\eta}$,

$$w_{S^{\text{shift}}}^{(j, r_j, p_j, v_j)}(t) = w_{\hat{S}}^{(j, \hat{r}_j, \hat{p}_j(1 + \frac{\eta^{\text{shift}}}{\beta(\hat{\mathcal{J}})}), \hat{v}_j(1 + \frac{\eta^{\text{shift}}}{\beta(\hat{\mathcal{J}})}))}(t - \bar{\eta}).$$

By definition of $\mathcal{J}_{\geq t_\lambda}^{\text{shift}}$, we also have that for all $(j, r_j, p_j, v_j) \in \mathcal{J}_{\geq t_\lambda}^{\text{shift}}$, $|r_j - \hat{r}_j| \leq \bar{\eta}$, $p_j \leq \hat{p}_j(1 + \frac{\eta^{\text{shift}}}{\beta(\hat{\mathcal{J}})})$ and $v_j \leq \hat{v}_j(1 + \frac{\eta^{\text{shift}}}{\beta(\hat{\mathcal{J}})})$. Hence, we can apply the second smoothness condition with $\mathcal{J}_1 = \mathcal{J}_{\geq t_\lambda}^{\text{shift}}$ and $\mathcal{J}_2 = \{(j, \hat{r}_j, \hat{p}_j(1 + \frac{\eta^{\text{shift}}}{\beta(\hat{\mathcal{J}})}), \hat{v}_j(1 + \frac{\eta^{\text{shift}}}{\beta(\hat{\mathcal{J}})})) : (j, r_j, p_j, v_j) \in \mathcal{J}_{\geq t_\lambda}^{\text{shift}}\} \subseteq \hat{\mathcal{J}}^{\text{up}}$. This gives:

$$
\begin{aligned}
F(S^{\text{shift}}, \mathcal{J}_{\geq t_\lambda}^{\text{shift}}) &\leq F(\hat{S}, \mathcal{J}_2) + \bar{\eta}\beta(\mathcal{J}_{\geq t_\lambda}^{\text{shift}})|\mathcal{J}_{\geq t_\lambda}^{\text{shift}}| \\
&\leq F(\hat{S}, \hat{\mathcal{J}}^{\text{up}}) + \bar{\eta}\beta(\mathcal{J}_{\geq t_\lambda}^{\text{shift}})|\mathcal{J}_{\geq t_\lambda}^{\text{shift}}| \\
&= F(\hat{S}, \hat{\mathcal{J}}^{\text{up}}) + \frac{\eta^{\text{shift}}}{\beta(\hat{\mathcal{J}})} \cdot \frac{\text{OPT}(\hat{\mathcal{J}})}{|\hat{\mathcal{J}}|} \cdot \beta(\mathcal{J}_{\geq t_\lambda}^{\text{shift}})|\mathcal{J}_{\geq t_\lambda}^{\text{shift}}| \\
&\leq F(\hat{S}, \hat{\mathcal{J}}^{\text{up}}) + \frac{\eta^{\text{shift}}}{\beta(\hat{\mathcal{J}})} \cdot \frac{\text{OPT}(\hat{\mathcal{J}})}{|\hat{\mathcal{J}}|} \cdot \beta(\hat{\mathcal{J}})\left(1 + \frac{\eta^{\text{shift}}}{\beta(\hat{\mathcal{J}})}\right)|\hat{\mathcal{J}}| \\
&= F(\hat{S}, \hat{\mathcal{J}}^{\text{up}}) + \eta^{\text{shift}}\left(1 + \frac{\eta^{\text{shift}}}{\beta(\hat{\mathcal{J}})}\right)\text{OPT}(\hat{\mathcal{J}}) \\
&\leq F(\hat{S}, \hat{\mathcal{J}}^{\text{up}}) + \eta^{\text{shift}}(1 + \eta^{\text{shift}})\text{OPT}(\hat{\mathcal{J}}), \quad (6)
\end{aligned}
$$

where the first inequality is by the second smoothness condition, the second one is is by monotonicity of $F$, the equality is by definition of $\bar{\eta}$, and the third inequality is by the smoothness and monotonicity of $\beta$. The last inequality is since $\beta \geq 1$.

Therefore, we get

$$
\begin{aligned}
\text{cost}(S^{\text{shift}}, \mathcal{J}_{\geq t_\lambda}^{\text{shift}}) &= E(S^{\text{shift}}) + F(S^{\text{shift}}, \mathcal{J}_{\geq t_\lambda}^{\text{shift}}) \\
&\leq E(\hat{S}) + F(\hat{S}, \hat{\mathcal{J}}^{\text{up}}) + \eta^{\text{shift}}(1 + \eta^{\text{shift}})\text{OPT}(\hat{\mathcal{J}}) \\
&= \text{OPT}(\hat{\mathcal{J}}^{\text{up}}) + \eta^{\text{shift}}(1 + \eta^{\text{shift}})\text{OPT}(\hat{\mathcal{J}}) \\
&\leq \left(1 + \beta(\hat{\mathcal{J}}^{\text{up}})\frac{\eta^{\text{shift}}}{\beta(\hat{\mathcal{J}})}\right)\text{OPT}(\hat{\mathcal{J}}) + \eta^{\text{shift}}(1 + \eta^{\text{shift}})\text{OPT}(\hat{\mathcal{J}}) \\
&\leq \left(1 + \beta(\hat{\mathcal{J}})\left(1 + \frac{\eta^{\text{shift}}}{\beta(\hat{\mathcal{J}})}\right)\frac{\eta^{\text{shift}}}{\beta(\hat{\mathcal{J}})}\right)\text{OPT}(\hat{\mathcal{J}}) + \eta^{\text{shift}}(1 + \eta^{\text{shift}})\text{OPT}(\hat{\mathcal{J}}) \\
&\leq (1 + \eta^{\text{shift}}(1 + \eta^{\text{shift}}))\text{OPT}(\hat{\mathcal{J}}) + \eta^{\text{shift}}(1 + \eta^{\text{shift}})\text{OPT}(\hat{\mathcal{J}}) \\
&= (1 + 2\eta^{\text{shift}}(1 + \eta^{\text{shift}}))\text{OPT}(\hat{\mathcal{J}}),
\end{aligned}
$$

where the first inequality is by (5) and (6), the second inequality is by the first smoothness condition with $\mathcal{J}_1 = \hat{\mathcal{J}}^{\text{up}}$ and $\mathcal{J}_2 = \hat{\mathcal{J}}$, the third inequality is by smoothness of $\beta$ and the last inequality since $\beta \geq 1$.

$\square$

**Corollary C.2.** *Assume that $\text{cost}(., .)$ is $\beta$-smooth, then*

$$OPT(\mathcal{J}^{\text{shift}}) \geq \left[\left(1 - \eta_2^{\frac{1}{\alpha}}\right)^\alpha - \eta^{\text{shift}}\right] \Big/ (1 + \eta^{\text{shift}})OPT(\hat{\mathcal{J}}).$$

*Proof.* We prove the result by contradiction. Assume that $\texttt{OPT}(\mathcal{J}^{\text{shift}}) < \left[ \left(1 - \eta_2^{\frac{1}{\alpha}}\right)^\alpha - \eta^{\text{shift}} \right] \Big/ (1 + \eta^{\text{shift}}) \texttt{OPT}(\hat{\mathcal{J}})$. Then, we have:

$$
\begin{aligned}
\texttt{OPT}(\hat{\mathcal{J}}^{\text{shift}}) &\leq \left( 1 + \beta(\hat{\mathcal{J}}^{\text{shift}}) \cdot \frac{\eta^{\text{shift}}}{\beta(\hat{\mathcal{J}})} \right) \texttt{OPT}(\mathcal{J}^{\text{shift}}) + \beta(\hat{\mathcal{J}}^{\text{shift}})|\hat{\mathcal{J}}^{\text{shift}}| \cdot \frac{\eta^{\text{shift}}}{\beta(\hat{\mathcal{J}})} \cdot \frac{\texttt{OPT}(\hat{\mathcal{J}})}{|\hat{\mathcal{J}}|} \\
&\leq \left( 1 + \beta(\hat{\mathcal{J}}) \cdot \frac{\eta^{\text{shift}}}{\beta(\hat{\mathcal{J}})} \right) \texttt{OPT}(\mathcal{J}^{\text{shift}}) + \beta(\hat{\mathcal{J}})|\hat{\mathcal{J}}^{\text{shift}}| \cdot \frac{\eta^{\text{shift}}}{\beta(\hat{\mathcal{J}})} \cdot \frac{\texttt{OPT}(\hat{\mathcal{J}})}{|\hat{\mathcal{J}}|} \\
&\leq (1 + \eta^{\text{shift}})\texttt{OPT}(\mathcal{J}^{\text{shift}}) + \eta^{\text{shift}}\texttt{OPT}(\hat{\mathcal{J}}) \\
&< \left( 1 - \eta_2^{\frac{1}{\alpha}} \right)^\alpha \texttt{OPT}(\hat{\mathcal{J}}).
\end{aligned}
$$

where the first inequality is by the first smoothness condition with $\mathcal{J}_1 = \hat{\mathcal{J}}^{\text{shift}}$, $\mathcal{J}_2 = \mathcal{J}^{\text{shift}}$, and the second one is by monotonicity of $\beta$.

Next, by definition of the error $\eta_2$, we have $\texttt{OPT}(\hat{\mathcal{J}} \setminus \hat{\mathcal{J}}^{\text{shift}}) = \eta_2 \cdot \texttt{OPT}(\hat{\mathcal{J}})$. Hence, by Lemma 3.1, there exists a schedule $S$ for $\hat{\mathcal{J}}^{\text{shift}} \cup (\hat{\mathcal{J}} \setminus \hat{\mathcal{J}}^{\text{shift}}) = \hat{\mathcal{J}}$ such that

$$
\begin{aligned}
\text{cost}(S, \hat{\mathcal{J}}) &\leq \left( \texttt{OPT}(\hat{\mathcal{J}} \setminus \hat{\mathcal{J}}^{\text{shift}})^{\frac{1}{\alpha}} + \texttt{OPT}(\hat{\mathcal{J}}^{\text{shift}})^{\frac{1}{\alpha}} \right)^\alpha \\
&< \left( (\eta_2 \texttt{OPT}(\hat{\mathcal{J}}))^{\frac{1}{\alpha}} + \left( \left( 1 - \eta_2^{\frac{1}{\alpha}} \right)^\alpha \texttt{OPT}(\hat{\mathcal{J}}) \right)^{\frac{1}{\alpha}} \right)^\alpha \\
&= \texttt{OPT}(\hat{\mathcal{J}}),
\end{aligned}
$$

which contradicts the definition of $\texttt{OPT}(\hat{\mathcal{J}})$. $\qquad\square$

**Corollary C.3.** *Assume that $\text{cost}(.,.)$ is $\beta$-smooth. If $\texttt{OPT}(\mathcal{J}) \leq \lambda \texttt{OPT}(\hat{\mathcal{J}})$, then $\eta_2 \geq \left( 1 - (\lambda(1 + \eta^{\text{shift}}) + \eta^{\text{shift}})^{\frac{1}{\alpha}} \right)^\alpha$.*

*Proof.* Assume that $\texttt{OPT}(\mathcal{J}) \leq \lambda \texttt{OPT}(\hat{\mathcal{J}})$. Since $\mathcal{J}^{\text{shift}} \subseteq \mathcal{J}$, we get $\texttt{OPT}(\mathcal{J}^{\text{shift}}) \leq \lambda \texttt{OPT}(\hat{\mathcal{J}})$. Hence, we have:

$$
\begin{aligned}
\texttt{OPT}(\hat{\mathcal{J}}^{\text{shift}}) &\leq \left( 1 + \beta(\hat{\mathcal{J}}^{\text{shift}}) \cdot \frac{\eta^{\text{shift}}}{\beta(\hat{\mathcal{J}})} \right) \texttt{OPT}(\mathcal{J}^{\text{shift}}) + \beta(\hat{\mathcal{J}}^{\text{shift}})|\hat{\mathcal{J}}^{\text{shift}}| \cdot \frac{\eta^{\text{shift}}}{\beta(\hat{\mathcal{J}})} \cdot \frac{\texttt{OPT}(\hat{\mathcal{J}})}{|\hat{\mathcal{J}}|} \\
&\leq \left( 1 + \beta(\hat{\mathcal{J}}) \cdot \frac{\eta^{\text{shift}}}{\beta(\hat{\mathcal{J}})} \right) \texttt{OPT}(\mathcal{J}^{\text{shift}}) + \beta(\hat{\mathcal{J}})|\hat{\mathcal{J}}^{\text{shift}}| \cdot \frac{\eta^{\text{shift}}}{\beta(\hat{\mathcal{J}})} \cdot \frac{\texttt{OPT}(\hat{\mathcal{J}})}{|\hat{\mathcal{J}}|} \\
&\leq (1 + \eta^{\text{shift}})\texttt{OPT}(\mathcal{J}^{\text{shift}}) + \eta^{\text{shift}}\texttt{OPT}(\hat{\mathcal{J}}) \\
&\leq (\lambda(1 + \eta^{\text{shift}}) + \eta^{\text{shift}})\texttt{OPT}(\hat{\mathcal{J}}).
\end{aligned}
$$

where the first inequality is by the first smoothness condition with $\mathcal{J}_1 = \hat{\mathcal{J}}^{\text{shift}}$, $\mathcal{J}_2 = \mathcal{J}^{\text{shift}}$, and the second one is by monotonicity of $\beta$.

Next, assume by contradiction, that $\eta_2 < \left( 1 - (\lambda(1 + \eta^{\text{shift}}) + \eta^{\text{shift}})^{\frac{1}{\alpha}} \right)^\alpha$, which implies that $\texttt{OPT}(\hat{\mathcal{J}} \setminus \hat{\mathcal{J}}^{\text{shift}}) < \left( 1 - (\lambda(1 + \eta^{\text{shift}}) + \eta^{\text{shift}})^{\frac{1}{\alpha}} \right)^\alpha \texttt{OPT}(\hat{\mathcal{J}})$.

Then, by Lemma 3.1, there exists a schedule $S$ for $\hat{\mathcal{J}}^{\text{shift}} \cup (\hat{\mathcal{J}} \setminus \hat{\mathcal{J}}^{\text{shift}}) = \hat{\mathcal{J}}$ such that

$$
\begin{aligned}
&\text{cost}(S, \hat{\mathcal{J}}) \\
&\leq \left( \texttt{OPT}(\hat{\mathcal{J}} \setminus \hat{\mathcal{J}}^{\text{shift}})^{\frac{1}{\alpha}} + \texttt{OPT}(\hat{\mathcal{J}}^{\text{shift}})^{\frac{1}{\alpha}} \right)^\alpha \\
&< \left( ((\lambda(1 + \eta^{\text{shift}}) + \eta^{\text{shift}})\texttt{OPT}(\hat{\mathcal{J}}))^{\frac{1}{\alpha}} + \left( \left( 1 - (\lambda(1 + \eta^{\text{shift}}) + \eta^{\text{shift}})^{\frac{1}{\alpha}} \right)^\alpha \texttt{OPT}(\hat{\mathcal{J}}) \right)^{\frac{1}{\alpha}} \right)^\alpha \\
&= \texttt{OPT}(\hat{\mathcal{J}}),
\end{aligned}
$$

which contradicts the definition of $\mathtt{OPT}(\hat{\mathcal{J}})$. Hence, $\eta_2 \geq \left(1 - (\lambda(1 + \eta^{\text{shift}}) + \eta^{\text{shift}})^{\frac{1}{\alpha}}\right)^{\alpha}$. $\qquad\square$

**Theorem C.4.** *Assume that cost$(.,.)$ is $\beta$-smooth. Then, for any $\lambda \in (0, 1], \eta^{shift} \in [0, 1)$, the competitive ratio of* TPE-S *run with trust parameter $\lambda$, a $\gamma$-competitive algorithm* ONLINEALG, *an optimal offline algorithm* OFFLINEALG, *and shift tolerance $\eta^{shift}$ is at most*

$$
\begin{cases}
\gamma & \text{if } OPT(\mathcal{J}) \leq \lambda OPT(\hat{\mathcal{J}}) \\[2mm]
\dfrac{\left((1+2\eta^{shift}(1+\eta^{shift}))^{\frac{1}{\alpha}}+\gamma^{\frac{1}{\alpha}}\left(\lambda^{\frac{1}{\alpha}}+\eta_1^{\frac{1}{\alpha}}\right)\right)^{\alpha}}{\max\left\{\lambda,\eta_1+\left(\left(1-\eta_2^{\frac{1}{\alpha}}\right)^{\alpha}-\eta^{shift}\right)(1+\eta^{shift})^{-1}\right\}} & \text{otherwise.}
\end{cases}
$$

*Proof.* Similarly as in the proof of Theorem 3.4, we have that if $\mathtt{OPT}(\mathcal{J}) \leq \lambda \mathtt{OPT}(\hat{\mathcal{J}})$, then

$$\text{cost}(S, \mathcal{J}) \leq \gamma \cdot \mathtt{OPT}(\mathcal{J}). \tag{7}$$

Next, we assume that there is $t_\lambda \geq 0$ such that $\mathtt{OPT}(\mathcal{J}_{\leq t_\lambda}) > \lambda \mathtt{OPT}(\hat{\mathcal{J}})$. Hence we immediately have:

$$\mathtt{OPT}(\mathcal{J}) \geq \mathtt{OPT}(\mathcal{J}_{\leq t_\lambda}) > \lambda \mathtt{OPT}(\hat{\mathcal{J}}).$$

By Corollary C.2 and by definition of the error $\eta_1$, we also get the following lower bound on the optimal schedule:

$$
\begin{aligned}
\mathtt{OPT}(\mathcal{J}) &\geq \mathtt{OPT}(\mathcal{J} \setminus \mathcal{J}^{\text{shift}}) + \mathtt{OPT}(\mathcal{J}^{\text{shift}}) \\
&\geq \eta_1 \mathtt{OPT}(\hat{\mathcal{J}}) + \left[\left(1 - \eta_2^{\frac{1}{\alpha}}\right)^{\alpha} - \eta^{\text{shift}}\right] \Big/ (1 + \eta^{\text{shift}}) \mathtt{OPT}(\hat{\mathcal{J}}).
\end{aligned}
$$

Therefore,

$$\mathtt{OPT}(\mathcal{J}) \geq \max\left\{\lambda, \eta_1 + \left[\left(1 - \eta_2^{\frac{1}{\alpha}}\right)^{\alpha} - \eta^{\text{shift}}\right] \Big/ (1 + \eta^{\text{shift}})\right\} \mathtt{OPT}(\hat{\mathcal{J}}). \tag{8}$$

We now upper bound the cost of the schedule output by our algorithm. By the same argument as in the proof of Lemma 3.2, we get:

$$\text{cost}(S^{on}, \mathcal{J} \setminus \mathcal{J}^{\text{shift}}_{\geq t_\lambda}) \leq \gamma \cdot \mathtt{OPT}(\hat{\mathcal{J}}) \left(\lambda^{\frac{1}{\alpha}} + \eta_1^{\frac{1}{\alpha}}\right)^{\alpha}.$$

Now, from Lemma C.1, we have

$$\text{cost}(S^{\text{shift}}, \mathcal{J}^{\text{shift}}_{\geq t_\lambda}) \leq (1 + 2\eta^{\text{shift}}(1 + \eta^{\text{shift}}))\mathtt{OPT}(\hat{\mathcal{J}}).$$

Therefore, by applying Lemma 3.1, we get:

$$
\begin{aligned}
\text{cost}(S, \mathcal{J}) &\leq \left(\text{cost}(S^{\text{shift}}, \mathcal{J}^{\text{shift}}_{\geq t_\lambda})^{\frac{1}{\alpha}} + \text{cost}(S^{on}, \mathcal{J} \setminus S^{\text{shift}})^{\frac{1}{\alpha}}\right)^{\alpha} \\
&\leq \mathtt{OPT}(\hat{\mathcal{J}}) \left((1 + 2\eta^{\text{shift}}(1 + \eta^{\text{shift}}))^{\frac{1}{\alpha}} + \gamma^{\frac{1}{\alpha}}(\lambda^{\frac{1}{\alpha}} + \eta_1^{\frac{1}{\alpha}})\right)^{\alpha}. \tag{9}
\end{aligned}
$$

Hence, we get the following upper bound on the competitive ratio of Algorithm 2:

$$\frac{\text{cost}(S, \mathcal{J})}{\mathtt{OPT}(\mathcal{J})}$$

$$= \mathbf{1}_{\mathtt{OPT}(\mathcal{J}) \leq \lambda \mathtt{OPT}(\hat{\mathcal{J}})} \frac{\text{cost}(S, \mathcal{J})}{\mathtt{OPT}(\mathcal{J})} + \mathbf{1}_{\mathtt{OPT}(\mathcal{J}) > \lambda \mathtt{OPT}(\hat{\mathcal{J}})} \frac{\text{cost}(S, \mathcal{J})}{\mathtt{OPT}(\mathcal{J})}$$

$$\leq \mathbf{1}_{\mathtt{OPT}(\mathcal{J}) \leq \lambda \mathtt{OPT}(\hat{\mathcal{J}})} \cdot \gamma + \mathbf{1}_{\mathtt{OPT}(\mathcal{J}) > \lambda \mathtt{OPT}(\hat{\mathcal{J}})} \frac{\left((1 + 2\eta^{\text{shift}}(1 + \eta^{\text{shift}}))^{\frac{1}{\alpha}} + \gamma^{\frac{1}{\alpha}}(\lambda^{\frac{1}{\alpha}} + \eta_1^{\frac{1}{\alpha}})\right)^{\alpha}}{\max\left\{\lambda, \eta_1 + \left[\left(1 - \eta_2^{\frac{1}{\alpha}}\right)^{\alpha} - \eta^{\text{shift}}\right] \Big/ (1 + \eta^{\text{shift}})\right\}}.$$

$$\square$$

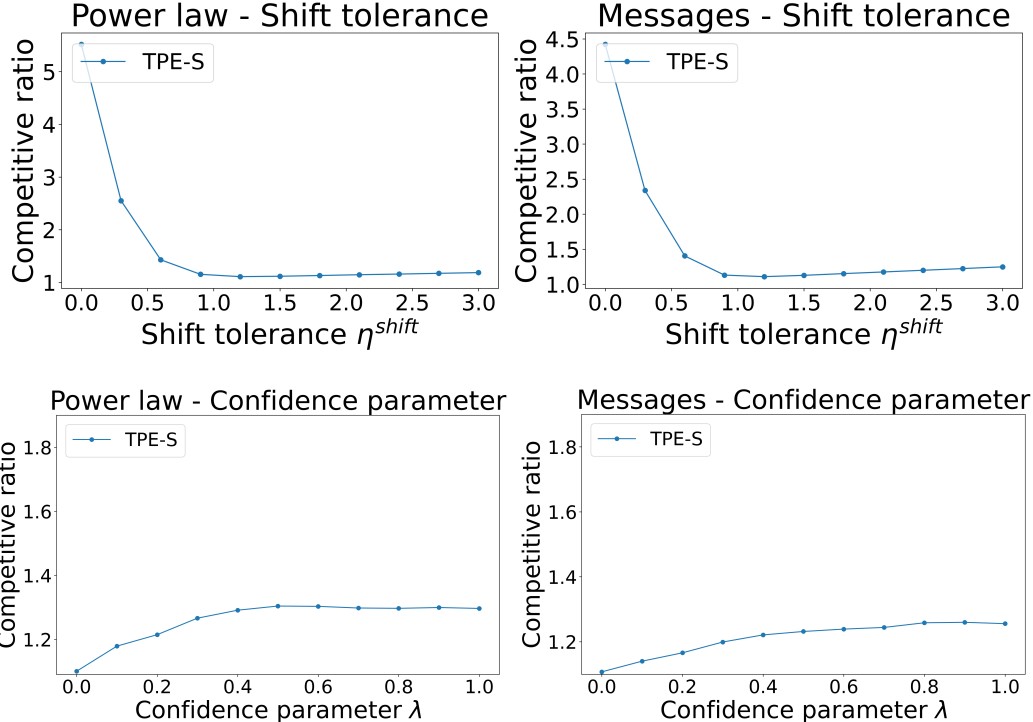

Figure 3: The competitive ratio achieved by our algorithm, TPE-S and the benchmark algorithm as a function of the shift tolerance $\eta^{shift}$ (row 1) and as a function of the confidence parameter $\lambda$ (row 2).

## E  Additional Experiments

Here, we also evaluate the impact of setting the parameters $\eta^{shift}$ and $\lambda$ on the two other datasets (power law and real datasets). The result are presented in Figure 3. We observe similar behaviors as for the periodic dataset.

## F  Comparison with [7, 4]

In [7, 4], the authors consider the energy minimization problem with deadlines, which, as detailed in Section 3.3, is a special case of our general framework. For this problem, they propose two different learning-augmented algorithms. We present here some elements of comparison with our algorithm for (GESP).

We first show in Section F.1 and F.2 that our prediction model and results generalize the ones in [7]: they are similar in the case of *uniform* deadlines and generalize the ones in [7] for *general* deadlines. We also note that they are incomparable to those in [4]. In Section F.3, we then discuss the algorithmic differences with [7] for the special case of energy with uniform deadlines.

### F.1  Discussion about the prediction and error model in comparison to [4, 7]

At a high level, the prediction models considered in [7] and [4] are qualified by [4] as 'orthogonal'. In [4], the number of jobs is known in advance, as well as the exact processing time for each job, however, the release time and deadlines are only revealed when a job arrives, and the error is proportional to the maximal shift in these values. On the contrary, in [7], the release times and deadlines are known in advance, and the prediction regards the total workload at each time step. The error is then defined as a function of the total variation of workload, which is the analog of additional and missing jobs in our setting.

Note that in the model in [4], the predicted and true set of jobs need to contain exactly the same number of jobs, whereas the model in [7] and our model allow for extra or missing jobs.

**Comparison with the prediction model and the error metrics in [7] for energy minimization with uniform deadlines.**

Note that the prediction model used in [7] for the energy minimization with deadlines problem is slightly different than ours: the prediction is the total workload $w_i^{\text{pred}}$ that arrives at each time step $i$ and needs to be scheduled before time $i + D$, and the error metric is defined as

$$\text{err}(w^{\text{real}}, w^{\text{pred}}) := \sum_i ||w_i^{\text{real}} - w_i^{\text{pred}}||^\alpha, \tag{10}$$

where $w^{\text{real}}$ denotes the real workload at each time step.

However, in the specific case of energy minimization under *uniform* deadline constraints, our prediction model and error metric and the ones from [7] are comparable: a workload $w_i^{\text{real}}$ that arrives at time $i$ is equivalent in our setting to receiving $w_i^{\text{real}}$ unit jobs with release time $r = i$ and a common deadline $d = i + D$. Moreover, we prove the following lemma, which shows that a small error in the sense of [7] induces a small error $\eta(\mathcal{J}, \hat{\mathcal{J}})$ in the sense defined in Section 2.

**Lemma F.1.** *For any constant $D > 0$, and any instance $(\mathcal{J}, \hat{\mathcal{J}})$, where at each time $i$, $\mathcal{J}$ is composed of $w_i^{\text{real}}$ jobs of one time unit with deadline $i + D$ and $\hat{\mathcal{J}}$ is composed of $w_i^{\text{pred}}$ jobs of one time unit with deadline $i + D$, we have:*

$$\eta(\mathcal{J}, \hat{\mathcal{J}}) \cdot \text{OPT}(\hat{\mathcal{J}}) = \max\{\text{OPT}(\mathcal{J} \setminus \hat{\mathcal{J}}), \text{OPT}(\hat{\mathcal{J}} \setminus \mathcal{J})\} \leq D \cdot \text{err}(w^{\text{real}}, w^{\text{pred}}).$$

*Proof.* For convenience, we write $\Delta_i = ||w_i^{\text{real}} - w_i^{\text{pred}}||$. We can then write $(\mathcal{J} \setminus \hat{\mathcal{J}}) \cup (\hat{\mathcal{J}} \setminus \mathcal{J})$ as the instance which, at each time step $i$, is composed of $\Delta_i$ unit size jobs with a common deadline $i + D$.

We now upper bound the optimal cost for $(\mathcal{J} \setminus \hat{\mathcal{J}}) \cup (\hat{\mathcal{J}} \setminus \mathcal{J})$ by the cost obtained by the Average Rate heuristic (AVR) (first introduced in [28]). For each $j$, The AVR algorithm schedules uniformly the $\Delta_j$ units of work arriving at time $j$ over the next $D$ time steps. This is equivalent to setting the speed $s_j(t)$ for each workload $\Delta_j$ at time $t \in [j, \ldots, j + D]$ to $\frac{\Delta_j}{D}$ and set $s_j(t) = 0$ everywhere else. For all $t \geq 0$, the machine then runs at total speed $\sum_j s_j(t)$.

Letting $E_{\text{AVR}}$ denote the total cost of the AVR heuristic, we get:

$$\max\{\text{OPT}(\mathcal{J} \setminus \hat{\mathcal{J}}), \text{OPT}(\hat{\mathcal{J}} \setminus \mathcal{J})\} \leq \text{OPT}((\mathcal{J} \setminus \hat{\mathcal{J}}) \cup (\hat{\mathcal{J}} \setminus \mathcal{J}))$$

$$\leq E_{\text{AVR}}((\mathcal{J} \setminus \hat{\mathcal{J}}) \cup (\hat{\mathcal{J}} \setminus \mathcal{J}))$$

$$= \sum_{t=1}^{\infty} \left( \sum_j \mathbf{1}_{s_j(t) \neq 0} s_j(t) \right)^\alpha$$

$$\leq \sum_{t=1}^{\infty} |\{j : s_j(t) \neq 0\}|^\alpha \left( \max_{j : s_j(t) \neq 0} s_j(t) \right)^\alpha$$

$$\leq \sum_{t=1}^{\infty} D^\alpha \left( \max_{j : s_j(t) \neq 0} s_j(t) \right)^\alpha$$

$$\leq \sum_{t=1}^{\infty} D^\alpha \sum_{j : s_j(t) \neq 0} s_j(t)^\alpha$$

$$= \sum_j D^\alpha s_j(t)^\alpha \sum_t \mathbf{1}_{s_j(t) \neq 0}$$

$$\leq \sum_j D^\alpha s_j(t)^\alpha D$$

$$= \sum_j \Delta_j^\alpha D$$

$$= D \cdot \mathrm{err}(w^{\mathrm{real}}, w^{\mathrm{pred}}),$$

where the fourth and sixth inequalities are since by definition of the AVR algorithm, each workload $\Delta_j > 0$ has only positive speed on time steps $[j, \ldots, j + D]$. $\qquad\square$

**Comparison of the error metrics for general objective functions.** We illustrate here that for a more general GESP problem, the error metric we define can be tighter than the one in [7] (in the sense that there are instances $(\mathcal{J}, \hat{\mathcal{J}})$ and quality cost functions $F$ such that $\eta(\mathcal{J}, \hat{\mathcal{J}}) << \mathrm{err}(w^{\mathrm{real}}, w^{\mathrm{pred}})$) and that it may better adapt to the specific cost function under consideration.

To illustrate this point, consider an instance where the prediction is the realization plus an additional workload of $k$ jobs that all arrive at time 0, and consider the objective of minimizing total energy plus flow time. In this case, the error computed in (10) is $k^\alpha$, whereas the error $\eta(\mathcal{J}, \hat{\mathcal{J}})$ we define is the optimal cost for the $k$ extra jobs. By using results from [3], this is equal to $k^{\frac{2\alpha-1}{\alpha}}$ ($<< k^\alpha$ when $\alpha$ grows large). Hence our error metric is tighter in this case.

### F.2 Comparison with the theoretical guarantees in [7]

We compare below the theoretical guarantees in Theorem 3.4 and the ones shown in [7] for the specific problem of energy minimization with *uniform* deadlines.

We note that we also generalize these results to the case of *general* deadlines to obtain the first guarantee that smoothly degrades as a function of the prediction error in that setting. Note that for general deadlines, [7] only obtain consistency and robustness, but not smoothness.

**Comparison in the case of uniform deadlines.** For convenience of the reader, we first recall below the guarantee proven in [7].

**Theorem F.2** (Theorem 8 in [7]). *For any given $\epsilon > 0$, algorithm LAS constructs, for the energy minimization with deadlines problem, a schedule of cost at most $\min\{(1 + \epsilon)\mathtt{OPT} + O((\frac{\alpha}{\epsilon})^\alpha)err, O((\frac{\alpha}{\epsilon})^\alpha)\mathtt{OPT}\}$, where*

$$err(w^{real}, w^{pred}) := \sum_i ||w_i^{real} - w_i^{pred}||^\alpha,$$

which is a similar dependency in $\epsilon$ as the one proved in Theorem 3.4. In particular, for all $\epsilon > 0$, Algorithm LAS achieves a consistency of $(1 + \epsilon)$ for a robustness factor of $O((\frac{\alpha}{\epsilon})^\alpha)$. On the other hand, when running Algorithm 1 with parameter $\lambda = (\frac{\epsilon}{C2^\alpha})^\alpha$ and the AVERAGE RATE heuristic [28] as ONLINEALG (which was proven to have a $2^\alpha$ competitive ratio in [7]), we obtain, by plugging $\lambda = (\frac{\epsilon}{C2^\alpha})^\alpha$ in the bounds provided in Corollary 3.5, a consistency of $(1 + \epsilon)$ for a robustness factor of $O(\frac{4^{\alpha^2}}{\epsilon^{\alpha-1}})$.

### F.3 Comparison with the algorithm (LAS) in [7]

In this section, we discuss the technical differences with the algorithm (LAS) proposed in [7] for the energy with deadlines problem.

We first note that [7] only shows smoothness, consistency and robustness in the uniform deadline case, where all jobs must be completed within $D$ time steps from their release time. For the general deadline case, [7] presents a more complicated algorithm and only show consistency and robustness. The authors note that "one can also define smooth algorithms for general deadlines as [they] did in the uniform case. However, the prediction model and the measure of error quickly get complex and notation heavy". On the contrary, Algorithm 1 remains simple, captures the general deadline case, and is also endowed with smoothness guarantees.

We now discuss more specifically the technical differences.

**Robustification technique.** [7] uses a convolution technique for the uniform deadline case, and a more complicated procedure that separates each interval into a base part and an auxiliary part for

the general deadline case. On the other hand, our robustification technique is based on a simpler two-phase algorithm.

We now give some intuition about why a direct generalisation of the techniques in [7] to general objective functions does not seem straightforward. The main technical difficulty is that in [7], each job $j$ must be completed before its deadline $d_j$, which is revealed to the decision maker at the time the job arrives and is used by the algorithm. For a general objective function, we do not have a deadline ; however, one could think about using the total completion time $c_j$ of each job instead. The issue is that $c_j$ may depend on all future job arrivals and is not known at the time the job arrives, hence it cannot be used directly by the algorithm.

To illustrate this point, consider the objective of minimizing total energy plus flow time with $\alpha = 2$. Consider two instances, where the first one has 1 job arriving at time 0 and the second one has 1 job arriving at time 0 and $n - 1$ jobs arriving at time $\frac{1}{\sqrt{n}}$. In the first case, the optimal is to complete the first job in 1 unit of time, whereas it is completed in $\frac{1}{\sqrt{n}}$ unit of time in the second case. Furthermore, this can be only deduced after the $n - 1$ other jobs have arrived. Hence, the completion times for the first job significantly differ in the two cases. Since it is not immediate how to generalize the technique in [7] without knowing $c_j$ at the time each job $j$ arrives, this motivated our choice of a different robustification technique.

**Smoothness and consistency technique.** To obtain smoothness and consistency guarantees, we use a similar technique as in [7] (summing the speeds obtained by computing an offline schedule for the predicted jobs and an online schedule for the extra jobs), with two main differences:
(1) In [7], the extra jobs arriving at each time $i$ are scheduled uniformly over the next $D$ time units. On the other hand, our algorithm computes the speeds for all extra jobs by following an auxiliary online algorithm given as an input to the decision maker. In fact, the technique from [7] can be interpreted as a special case of our algorithm, where the auxiliary algorithm is the AVERAGE RATE heuristic [28].
(2) The offline schedule we compute is conceptually identical to the one used in [7], however, our online schedule differs, as it needs to integrate two different types of extra jobs: (1) the extra jobs that arrive during the second phase of the algorithm ($t \geq t_\lambda$), and (2) the jobs that were not finished during the first phase of the algorithm. [7] only needs to handle the first type of extra jobs. This results in a different analysis.