# OpenReview forum: "Energy-Efficient Scheduling with Predictions"
_NeurIPS.cc/2023/Conference — NeurIPS 2023 poster_

### Official Review · Reviewer_HbG7 · 2023-07-06

**Soundness:** 3 good
**Presentation:** 3 good
**Contribution:** 3 good
**Rating:** 6
**Confidence:** 4

**Summary:**

This paper studies an energy-efficient scheduling problem under the setting of prediction. In this problem, each job has a release time and processing time. The job arrives online and the algorithm can determine the speed of the machine. The higher speed means a higher energy cost. The total energy cost integrates the speed cost over all time points. Besides the energy cost, the objective contains a job cost function that relies on the schedule. In this paper, the authors assume this cost function is subadditive to capture more applications.
The schedule is defined as the speed of the machine each time. The goal is to find a feasible schedule with the minimum total cost.

This paper considers the above problem under the setting of prediction, where their prediction is the release time and processing time of each job. The main contribution of this work is a learning-augmented algorithm that achieves (1+\epsilon) consistency and bounded robustness. The authors also give the ratio function depending on the prediction error. Finally, the authors verify the performance of the proposed algorithm in some real datasets.


**Strengths:**

The submission is carefully written and structured, so reads well given the technicality of the material. Especially, the authors provide sufficient intuitions to help understand the algorithm.
The proposed learning-augmented algorithm can be applied to several problems. Although one still needs to utilize the classical online and offline algorithms for each specific problem, the proposed framework provides some intuitions to show how to combine these two algorithms.


**Weaknesses:**

I am one of the reviewers of the previous version of this paper. My previous major concerns are: (1) the authors didn’t provide a comparison of the proposed framework and the existing works, thus it is unclear how good this framework is if we apply it to some specific problem; (2) the considered problem is too general and may not admit a polynomial time algorithm.

In this version, the author addresses these two points appropriately. For the first point, the authors added Table 1 to discuss four related special cases. For the second point, the authors considered the approximation ratio in the proof of Theorem 3.4.


**Questions:**

I would appreciate it if the authors add a discussion about how to compute the error function given two instances.

**Limitations:**

This is a theoretical paper, there is no potential negative societal impact.

---

> ### Author Rebuttal · Authors · 2023-08-09
>
> We thank the reviewer for their careful and positive review.
>
> $\bullet$ “I am one of the reviewers of the previous version of this paper. My previous major concerns are […] In this version, the author addresses these two points appropriately.”\
> We are happy to hear that the previous weaknesses were addressed appropriately.
>
> $\bullet$ “I would appreciate it if the authors add a discussion about how to compute the error function given two instances.”\
> First, we would like to emphasize that our algorithm does not require knowledge of the prediction error $\eta$. Nevertheless, if one wanted to compute this prediction error, they would have to compute the optimal offline cost OPT of three different instances. There are previous works that have studied the offline version of energy-efficient scheduling. In particular, [28] gives an optimal offline algorithm for the energy minimization with deadlines problem and [2] gives an optimal offline algorithm for the energy plus flow time problem in the case of unweighted unit-size jobs.

---

> > ### Comment · Reviewer_HbG7 · 2023-08-18
> >
> > I understand that the algorithm does not require to compute the error function. But, my point is that the error function is important for training predictors, and it can be viewed as an initial version of the loss function. Besides this, the error function can also be used to mark the good and bad predictions during the training process. So it is useful to give an algorithm to compute the error function.

---

> > > ### Author Response · Authors · 2023-08-20
> > >
> > > We agree with the reviewer that computing the prediction error can be helpful during the training process. We will include a discussion about computing the prediction error in the paper.

---

### Official Review · Reviewer_RAEY · 2023-07-06

**Soundness:** 3 good
**Presentation:** 3 good
**Contribution:** 3 good
**Rating:** 7
**Confidence:** 4

**Summary:**

The authors study energy-efficient scheduling with predictions. There
are already previous works on minimizing energy consumption in a setting
with deadlines. The authors provide a unified way to address both
the scheduling with deadlines as well as (weighted) flow time plus energy cost.
It was already known that no algorithm can be both 1-consistent and robust
and therefore one has to aim for a trade-off between consistency and robustness.
They provide consistency and robustness trade-offs which scale roughly with
(1/lambda), where lambda is the trade-off parameter. They also show a lower
bound on robustness scaling with sqrt(1+1/lambda), i.e., there is still a
space for improvement.
They also discuss deterioration of their bounds between consistency and
robustness as a function of a prediction error defined in a natural way,
especially with their extension to approximately correct predictions.
I did not notice any lower bounds on the dependence on prediction error,
also the formula stating the dependence is not easy to understand (maybe a plot
would help?). But overall, I find the contribution of this paper solid and
I will be happy to see it accepted.


**Strengths:**

they provide an unified framework for a broad range of problems in energy efficient scheduling

**Weaknesses:**

* resulting bounds for the Deadline setting are weaker than in previous works
* results not tight yet

**Questions:**

* you could elaborate a bit on how does your competitive ratio behave as a dependency on prediction error, if possible to give a clean answer.

**Limitations:**

* Authors stated their theoretical results formally, including all assumptions made

---

> ### Author Rebuttal · Authors · 2023-08-09
>
> We thank the reviewer for their careful and positive review.
>
> $\bullet$ “elaborate a bit on how does your competitive ratio behave as a dependency on prediction error”\
> Such a discussion is provided in Appendix G.1. We would be happy to move that discussion to the main body of the paper. A slightly edited version of this discussion is given below for convenience.
> The competitive ratio smoothly approaches the consistency bound as the prediction error tends to 0. In addition, it distinguishes the effect of two possible sources of errors:
>
> (1) when there is a growing number of predicted jobs that do not arrive ($\eta_1 = 0$ and $\eta_2$ goes to 1), the upper bound degrades monotonically to $O(1/\lambda)$.
>
> (2) when there is a growing number of unpredicted jobs that arrive online ($\eta_1$ goes to infinity and $\eta_2 = 0$), the competitive ratio first deteriorates and then improves, with an asymptotic rate equal to the competitive ratio $\gamma_{on}$ achieved by the online algorithm OnlineAlg. This behavior is because our algorithm mostly follows the online algorithm when the cost of the additional jobs dominates.

---

> > ### Comment · Reviewer_RAEY · 2023-08-12
> >
> > thank you for your explanation.

---

### Official Review · Reviewer_BEvj · 2023-07-06

**Soundness:** 4 excellent
**Presentation:** 3 good
**Contribution:** 2 fair
**Rating:** 5
**Confidence:** 4

**Summary:**

The paper considers speed scaling scheduling with learning augmented predictions. In contrast to previous works that considered the deadline-based version of the problem, the current paper studies a more general model that allows for different quality of service objectives to be optimised alongside the energy consumption of the schedule. These include for example the well studied total flow time objective.

Similarly to other results in the learning augmented algorithms literature, the employed approach is that of combining at runtime an offline and an online algorithm for the problem.

Finally, experimental results over real and sytnthetic datasets are provided.



**Strengths:**

The biggest strength of the paper is the generality of the considered model and how it implies algorithms with predictions for a number of different speed-scaling settings.

Other strengths include that the paper is well written, and obtains a clean and general framework.

**Weaknesses:**

The prediction error is quite unnatural in my personal opinion. But going for it is that it generalizes the prediction error considered in [7].

The major weakness is that jobs are considered to be predicted correctly if the prediction for the job is 100% accurate. This is a bit unnatural and it could be argued that if the predicted parameters of every job are very close to the real ones then the predictions are adequate and should be useful. This is tackled in Section 4 (and mainly Appendix C) but to my understanding it only allows for small errors per job -- which is a weakness compared to other papers in the area in general and specifically previous papers on speed scaling with predictions.

The technique employed in the paper is quite standard and I couldn't identify any really new idea. Nevertheless the results are not obvious and had to be worked out.



**Questions:**

I am wondering if there is a way to simplify the competitive ratio formula in Theorem 3.4 -- even perhaps at a slight loss of the proven guarantee? I did check the discussion in the appendix which is helpful but still does not give a very clear picture of the obtained result.

**Limitations:**

No conceivable limitations or potential negative societal impact.

---

> ### Author Rebuttal · Authors · 2023-08-09
>
> We thank the reviewer for their careful review. We believe there is a misunderstanding regarding the major weakness raised by the reviewer, which we address first below. Please let us know if this is not the case, we would be happy to also answer any follow-up questions.
>
> $\bullet$ “This is tackled in Section 4 (and mainly Appendix C) but to my understanding it only allows for small errors per job -- which is a weakness compared to other papers in the area in general and specifically previous papers on speed scaling with predictions.” \
> The competitive ratio we achieve gracefully degrades as a function of the (per job) shift tolerance parameter $\eta^{shift}$, which is chosen by the algorithm designer depending on the instance. In particular, $\eta^{shift}$ can be set to be large to allow for large errors per job (at the cost of a worse competitive ratio). Thus, we disagree with the claim that our algorithm only allows for small errors per job.
>
> $\bullet$ “The technique employed in the paper is quite standard and I couldn't identify any really new idea. Nevertheless the results are not obvious and had to be worked out.”\
> We respectfully disagree with the claim that our techniques are standard. In particular, many algorithms with predictions first trust the predictions and then, if the cost becomes too large, switch to an online algorithm that ignores the predictions. Our approach is to instead first run an online algorithm that ignores the predictions and then switch to trusting the predictions. We are not aware of previous work that has used this approach. If the reviewer has a reference to a paper that uses such an approach, we would be grateful if they shared it.
>
> $\bullet$ “I am wondering if there is a way to simplify the competitive ratio formula in Theorem 3.4 -- even perhaps at a slight loss of the proven guarantee? I did check the discussion in the appendix which is helpful but still does not give a very clear picture of the obtained result.”\
> We attempted to further simplify the formula but, unfortunately, weren’t able to do so. We agree with the reviewer that the formula is complicated. We are happy to move some of the discussion about this formula from the appendix to the main body of the paper.

---

> > ### Comment · Reviewer_BEvj · 2023-08-11
> >
> > Thank you for the response and for clarifying that the algorithm allows for larger errors.
> >
> > I find it quite far-fetched to consider it a different technique whether one starts following the prediction or a robust algorithm or vice versa. And it is also not novel, many papers in the area combine an algorithm following the predictions and a robust algorithm in an "experts sense" so that they compare well versus the in hindsight best of them. If the robust algorithm has better cost at the beginning of the instance then the algorithm with predictions will start by following that. See "Online Metric Algorithms with Untrusted Predictions" for an example. Another class of problems where that can be the case is in cases where one cannot "recover" from a wrong decision. For instance in the secretary problem any reasonable algorithm with predictions would not hire the secretary at the early stages even if the predictions suggest so.
> >
> > My evaluation remains unchanged.

---

### Official Review · Reviewer_fguw · 2023-07-25

**Soundness:** 4 excellent
**Presentation:** 3 good
**Contribution:** 3 good
**Rating:** 6
**Confidence:** 3

**Summary:**

This paper adds to the literature on energy efficient scheduling by defining an algorithm that extends the problems of "energy minimization with deadlines" and "energy plus flow time minimization" to the case where predictions about future jobs are available. The paper assumes an existing algorithm for online and offline energy efficient scheduling (wrt to a particular objective function) as well as an algorithm that makes predictions. In this case they give consistency - how far cost of the with-predictions variant deviates from the without-predictions variant - and robustness - how bad things can get when predictions are very wrong - bounds for their algorithm and compare them to existing results in the literature.

They explore several other properties of these algorithms and give an extension of the algorithm to small deviations - i.e. the case where predictions don't have to be exactly correct to be considered "good".

Experiments are done on two synthetic and one real (SNAP College Message) dataset.


**Strengths:**

The paper tackles an important problem, where improved algorithms will have a positive environmental impact.

The paper contributes a new bound on a problem that hasn't been studied yet. It also adds insight into previously studied variants of the problem.

Relevant claims seem to be justified with proofs - although I was unable to fully check everything, but what I did check was consistent.


**Weaknesses:**

This paper is abstract and jargon / notation heavy and will be difficult for someone without deep familiarity with the problem domain to read. On place that I had trouble were understanding how speed is assigned to a job.

The variable lambda is prominently used, but is not given an intuitive definition until line 195. I would like to see a definition of this "confidence level" variable nearby where the intuitive definition of alpha is given.

Around line 196, I would like to see examples or citation of what some of these ONLINE and OFFLINE algorithms are, as well as an example of or reference to an algorithm for prediction.

Line 29: Replacing: "is equal to speed to some power" with "is equal to speed *raised* to some power" would make this easier to understand. It does make sense as-is, but the added clarification would have helped me.

Line 174: Using eta as a function and a numeric variable is confusing.


**Questions:**

I don't understand on line 162, prediction error is defined with an alpha-norm, but perhaps reading the relevant citation more carefully could help there.

I'm wondering about the assumption that the speed of the machine is equal to the sum of the speeds of all jobs currently executing. Is overhead from context switching / cache misses negligible in real life, or is this just a simplification for theoretical analysis?

Have the authors considered codifying their proofs in proof assistant software (e.g. Lean, Coq)?


**Limitations:**

The discussion is sufficient.

---

> ### Author Rebuttal · Authors · 2023-08-09
>
> We thank the reviewer for their careful and overall positive review. We hope the answers below address the reviewer’s concerns. Please let us know if this is not the case, we would be happy to also answer any follow-up questions.
>
> $\bullet$ “One place that I had trouble were understanding how speed is assigned to a job.”\
> The assignment of speed $s_j(t)$ to a job $j$ at time $t$ is the main technical part of our algorithm and it is indeed subtle as it depends on both $j$ and $t$. More precisely, if time t is during the first phase of the algorithm, then $s_j(t)$ is the speed according to the online algorithm OnlineAlg for all jobs $j$. If $t$ is during the second phase of the algorithm, then $s_j(t)$ is assigned according to the offline algorithm OfflineAlg for the correctly predicted jobs $j$ and it is assigned according to OnlineAlg for the incorrectly predicted jobs $j$. We will add some discussion in the description of the algorithm to clarify this.
>
> $\bullet$ “The variable lambda is prominently used, but is not given an intuitive definition until line 195.”\
> lambda is the parameter that controls the tradeoff between consistency and robustness in our results.  Intuitively, the more we trust the predictions, the smaller lambda should be chosen. On page 2, we define lambda as being a parameter such that our results hold “for any lambda in (0, 1]” (see caption of Table 1). We will clarify this in the next version of the paper.
>
> $\bullet$ “Around line 196, I would like to see examples or citation of what some of these ONLINE and OFFLINE algorithms are, as well as an example of or reference to an algorithm for prediction.”\
> In Section 3.3, we provide a discussion of such algorithms, as well as many references.  We would be happy to move some of that discussion to earlier in the paper. Regarding a “reference to an algorithm for prediction”, we provide several references in the introduction for relevant algorithms with predictions.
>
> $\bullet$ “I don't understand on line 162, prediction error is defined with an alpha-norm, but perhaps reading the relevant citation more carefully could help there.”\
> We emphasize that this prediction error on line 162 is not the prediction error we use in this paper but the one considered in [7], so we do not provide a detailed discussion of it. This other definition follows from the impossibility result shown in [7] (Appendix C).
>
> $\bullet$ “Is overhead from context switching / cache misses negligible in real life, or is this just a simplification for theoretical analysis?”\
> It is common in the scheduling literature (with or without predictions) to assume that context switching has a negligible cost and to ignore it. This assumption is justified in many, but not all computing environments. However, there are some environments that have a high cost for preemption (such as those that involve a physical process or those that depend upon a massive amount of data), and typically this is handled by not allowing any preemptions.  There have been some works that assign a cost to preemption and/or limit the number of preemptions. Our work does not handle these cases, but it is a good future research direction.
>
> $\bullet$ “Have the authors considered codifying their proofs in proof assistant software (e.g. Lean, Coq)?”\
> No, we have not. We strongly believe that using such software is not standard for papers in this area at NeurIPS.

---

> > ### Comment · Reviewer_fguw · 2023-08-17
> >
> > Thank you for the clarifications.
> >
> > > “Have the authors considered codifying their proofs in proof assistant software (e.g. Lean, Coq)?”
> > > No, we have not. We strongly believe that using such software is not standard for papers in this area at NeurIPS.
> >
> > I agree it is not a requirement. I still recommend looking into it as a way to increase the quality and interoperability of any theoretical work and remove margins of error.

---

> > > ### Author Response · Authors · 2023-08-20
> > >
> > > Thank you for the recommendation, we will look into it. Is there a particular software you recommend?

---

> > > > ### Comment · Reviewer_fguw · 2023-08-21
> > > >
> > > > I'm currently learning Lean4, but I'm not an expert in the subject. There are not that many of them: https://en.wikipedia.org/wiki/Proof_assistant and Coq and Lean - both of which have foundations in type theory - seem to be the frontrunners.
> > > >
> > > > However, I do believe these will become necessary in mathematical works as time progresses. It is far too easy for a human reviewer to make a mistake when reading the math. For proofs to be truely useful (i.e. re-usable) and trusted they need to be codified in some computational framework.

---

### Decision · Program_Chairs · 2023-09-21

**Decision:**

Accept (poster)

**Comment:**

The paper considers an important scheduling problem. The reviewers generally appreciated the technical results and the problem. The main critique is that the error model used could be considered unnatural. Still, the results are likely tobe of interest to the community.